# Uncovering low-frequency vibrations in surface-enhanced Raman of organic molecules

Alexandra Boehmke Amoruso[1], Roberto A. Boto [2,3], Eoin Elliot[1], Bart de Nijs [1], Ruben Esteban [2,3], Tamás Földes[4], Fernando Aguilar-Galindo [3,5,6], Edina Rosta [4], Javier Aizpurua [3,7,8] ✉ & Jeremy J. Baumberg [1] ✉

Accessing the terahertz (THz) spectral domain through surface-enhanced Raman spectroscopy (SERS) is challenging and opens up the study of low-frequency molecular and electronic excitations. Compared to direct THz probing of heterogenous ensembles, the extreme plasmonic confinement of visible light to deep sub-wavelength scales allows the study of hundreds or even single molecules. We show that self-assembled molecular monolayers of a set of simple aromatic thiols confined inside single-particle plasmonic nanocavities can be distinguished by their low-wavenumber spectral peaks below 200 cm$^{-1}$, after removal of a bosonic inelastic contribution and an exponential background from the spectrum. Developing environment-dependent density-functional-theory simulations of the metal-molecule configuration enables the assignment and classification of their THz vibrations as well as the identification of intermolecular coupling effects and of the influence of the gold surface configuration. Furthermore, we show dramatically narrower THz SERS spectra from individual molecules at picocavities, which indicates the possibility to study intrinsic vibrational properties beyond inhomogeneous broadening, further supporting the key role of local environment.

Infrared-energy vibrational spectra can fingerprint characteristic bonds identifying a material and are accessible with infrared absorption and Raman spectroscopies. Molecular vibrations at these frequencies (1000–3000 cm$^{-1}$) are typically localized to specific functional groups. Delocalised vibrations spanning the length of the molecule and mixing intra- and inter-molecular modes are found at lower energy (< 10 THz, 333 cm$^{-1}$, 41 meV, $\lambda$ > 30 μm), and provide both structural and environmental information. Low-frequency vibrational spectra have historically been less accessible due to a lack of laser sources, detectors, and filters[1]. Although technology now exists for absorption and Raman spectroscopies at these frequencies (generally referred to as terahertz, THz), they require large sample volumes, which leads to averaging over disorder, giving broad or overlapping spectra that require complex interpretation[2]. THz vibrational spectroscopies are utilized in organic electronics, biophysics, pharmaceuticals, and the food and explosives industries for their ability to monitor structural effects such as phase transitions in molecular solids[3], interlayer coupling in van-der-Waals materials[4], relaxation

[1]NanoPhotonics Centre, Cavendish Laboratory, J J Thomson Avenue, University of Cambridge, Cambridge, UK. [2]Centro de Física de Materiales CFM-MPC (CSIC UPV/EHU), Donostia-San Sebastián, Spain. [3]Donostia International Physics Center (DIPC), Donostia-San Sebastián, Spain. [4]Department of Physics and Astronomy, University College London, London, UK. [5]Institute for Advanced Research in Chemical Sciences (IAdCHEM), Universidad Autónoma de Madrid, Madrid, Spain. [6]Departamento de Química, Universidad Autónoma de Madrid, Madrid, Spain. [7]Ikerbasque, Basque Foundation for Science, Bilbao, Spain. [8]Dept. of Electricity and Electronics, University of the Basque Country (UPV/EHU), Leioa, Spain. ✉e-mail: aizpurua@ehu.eus; jjb12@cam.ac.uk

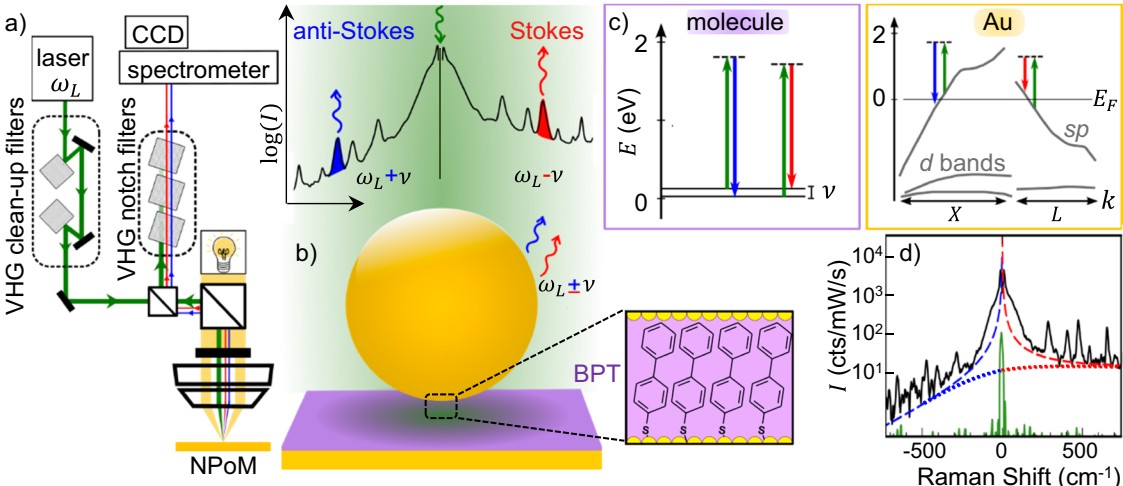

**Fig. 1 | THz SERS spectroscopy. a** Schematic optics with 785-nm laser pre-filtered by two volume holographic gratings (VHGs) and focused onto the sample by NA = 0.8 objective. Filtering the scattered light with three notch VHGs removes the laser before the spectrometer and CCD. **b** Au nanoparticle-on-mirror (NPoM) confines incident laser to nm-scale gap volume, producing anti-Stokes and Stokes Raman scattering at $\omega_L \pm \nu$. Inset: height of NPoM gap set by molecular monolayer (here biphenyl-4-thiol, BPT). **c** Vibrational (purple) and electronic (orange box) Raman scattering processes, colours as in (**a**, **b**). **d** Typical SERS from BPT-NPoM (black) compared to flat gold (green). Background fits show the expected background from the electronic Raman scattering model (see text, red/blue dashed lines) and Au PL model (red/blue dotted).

dynamics in amorphous solids[5] or hydrogen-bonded networks[6], and collective motion across biological macromolecules[7].

On much smaller nanoscale samples, the ability to observe low-frequency vibrations would enable monitoring of molecular conformations and attachments[8], intermolecular interactions[9], chemical steric effects[10], self-assembly of biological building blocks[11], and molecule-metal interfaces in electrochemical[12,13] or electronic devices[14–16]. To do so requires surface-enhanced Raman spectroscopy (SERS), as this technique can probe sub-wavelength volumes while enhancing the signal from a few emitters. Although SERS is a reliable vibrational spectroscopy[17], here we push it to its low-frequency limit, which is rarely utilised, to explore the THz modes of molecular monolayers using well-defined plasmonic constructs that enable probing down to single molecules through billion-fold signal enhancements[18]. THz SERS has been used in previous pioneering work to observe mass changes due to electrochemical reactions[13] and crystal-facet dependence of THz modes on molecules adsorbed onto Au surfaces[19], by a method developed in prior work[20]. Here we present an alternative approach which identifies the representative experimental molecular SERS spectrum in the THz and IR regions using large datasets where the background emission from the Au nanostructure is properly removed. Furthermore, we present single- (or few-) molecule observations and detailed DFT simulations which reveal the impact of the local environment and of intermolecular interactions on the THz modes.

## Results

By combining ultra-narrow volume holographic grating (VHG) notch filters with nanoparticle microscopy (Fig. 1a), single-particle SERS can simultaneously detect Stokes and anti-Stokes Raman scattering down to ± 5 cm$^{-1}$ Raman shifts emitted by the < 200 molecules within individual nanoparticle-on-mirror (NPoM) plasmonic nanocavities (Fig. 1b)[21,22]. These constructs embed a self-assembled monolayer (SAM) of molecules between a flat Au surface and a Au nanoparticle electrostatically deposited on top[23]. The plasmonic dimer-like cavity supports plasmonic modes with induced surface charges that couple across the SAM-defined nano-gap[24], forming a hotspot with enhanced optical field $E/E_0 \sim 300$ and billion-fold amplified SERS $\propto |E/E_0|^4$. By collecting large data sets on systematically optimised

systems, a statistically representative spectrum of any sample can be identified[19,25].

In this work, the low-energy SERS of a family of seven aromatic thiol molecules are measured and compared [Fig. 2, (1–7)]. The molecules vary in length and functional groups resulting in different effective mass, length, and elasticity. All are thiols because the S-Au bond anchors the formation of well-ordered SAMs[26]. The NPoMs are constructed with nanoparticles of ≈ 80 nm size and with a small range of gap thicknesses set by the SAM such that the strongest localized plasmon resonance $\omega_c$ overlaps with the laser frequency $\omega_L$. The robust nature of the NPoM construct means that despite the sensitivity of plasmonic confinement to the exact nano-scale and molecular architecture, the plasmon spectral peak is well constrained[22].

## Experimental results

By automating measurements to collect a series of SERS spectra from each of over a hundred NPoMs, as well as dark-field spectra to record each $\omega_c$ (Supplementary Fig. S7), representative spectra for each molecule are obtained (Fig. 2). For each molecule, the width of the plasmon peak distribution from all NPoMs characterizes the uniformity of the SAM, defining a standard for SAM quality in sample fabrication (Supplementary Fig. S7). This is crucial to set a well-defined NPoM gap, but also because THz vibrational modes depend strongly on molecular conformation and their local environment, since these modes consist of delocalized vibrations over a large number of atoms in the molecule, whereas higher-energy vibrations involve very localized motion of well-defined bonds, less affected by large scale conformation and their environment. For each molecule, the spectral positions of the plasmonic resonances are compatible with their molecular lengths and tilt angle (Supplementary Table S2), as calculated using a full quasinormal-mode analysis[27] of the NPoM geometry.

These representative THz SERS spectra consist of molecular vibrational peaks superimposed on an asymmetric background. This background rises steeply for frequencies close to that of the laser (Raman shifts $|\nu| < 200$ cm$^{-1}$), on both Stokes and anti-Stokes sides (black line Fig. 1d) but is completely absent in Raman spectra of molecular solids (Supplementary Fig. S1) and flat Au (so is not from laser scatter). While it is clear this background continuum arises as a consequence of the penetration of the NPoM plasmonic field into the

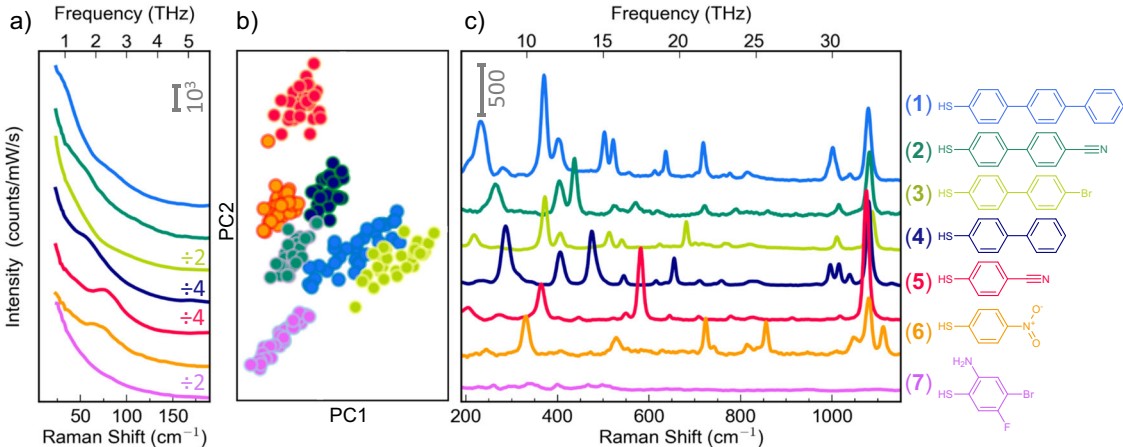

**Fig. 2 | Comparison of low-frequency modes. a, c** Low- and high-frequency SERS Stokes spectra of seven molecules forming SAMs inside 80 nm Au NPoMs. Representative spectra were selected from several hundred NPoMs (see text). ERS bosonic background is fit and subtracted by process detailed in Supplementary Note 2.

**b** Thirty low-frequency spectra of each molecule clustered and projected using PCA. The Marker area is coloured according to the true label of molecules (**1–7**), and the edges are coloured according to their cluster label (see text).

metal due to its tight optical confinement and enhanced coupling in the small gap[22], controversy remains in the literature about whether it arises from electronic Raman scattering (ERS) or Au photoluminescence (PL). In the former ERS mechanism (Fig. 1c orange box), plasmons excite electrons near the Fermi energy ($E_F$) to a virtual state (dashed) before relaxing to an empty state above/below $E_F$[28]. The other leading hypothesis is Au photoluminescence induced by intraband transitions, where plasmons excite hot electrons that recombine with carriers near $E_F$[29]. These background contributions are often arbitrarily fit with a polynomial and subtracted out[30], however the strong increase of the background signal in the THz range encourages us to select a specific fit function, appropriate to a physical model.

Here, the 785 nm ($\hbar\omega_L = 1.6$ eV) laser excites neither molecular HOMO-LUMO nor Au interband electronic transitions[31], while the localization length in the Au nanocavities should provide insufficient momentum for intraband PL transitions[32]. Furthermore, the frequency-dependence of the intraband PL intensity spectrum has recently been shown theoretically to follow that of the excited electron population, which has a Fermi-Dirac distribution[33] (Fig. 1d dotted). By contrast, the ERS intensity should depend on frequency according to a Bose-Einstein distribution ($n_{BE}$)[34], because the light inelastically scatters from (bosonic) quasiparticles of the electron gas, as widely discussed in the 1980s for understanding excitations in metals[35]. This indeed provides a good fit to the observed asymmetric background (Fig. 1d, dashed), following

$$I_{Au}(\nu) \propto \left|\text{EF}(\omega_L - \nu)\right|^2 \cdot \chi \cdot \left[n_{BE}(\nu) + \theta(\nu)\right], \quad (1)$$

where $\text{EF}(\omega_L - \nu)$ is the outcoupling efficiency (of optical field) enhanced around the plasmon frequency $\omega_c$, $n_{BE} = \left[\exp\{h\nu/k_B T\} - 1\right]^{-1}$ is the bosonic thermal population at energy $h\nu$, $\nu$ is the Raman shift, $\chi$ is the electronic Raman susceptibility of the metal, and $\theta$ is the Heaviside function, $\theta = 0$ ($\nu < 0$, anti-Stokes) or 1 ($\nu > 0$, Stokes)[36]. Here $\text{EF}(\omega)$ for each NPoM is modelled as a broad peak centred at the plasmon resonance $\omega_c$ (simultaneously recorded in all these experiments), and the incoupling efficiency $\left|\text{EF}(\omega_L)\right|^2$ is omitted in Eq.(1) as it is constant. We use a simulated-annealing algorithm to fit the measured SERS backgrounds to Eq.(1), in all cases giving electronic temperatures $T \sim 320 \pm 20$ K (see "Methods" and Supplementary Fig. S3). The close fit of Eq.(1) to the SERS background gives empirical evidence that the emission is ERS and not PL (Fig. 1d).

Removing these backgrounds (denoted the 'bosonic' background in the following) allows the molecules to be directly compared (Fig. 2).

It is evident that the low-frequency spectra still contain an exponential component $\propto \exp\{-\nu/\xi\}$ (Fig. 2a). We fit and remove this additional background, finding a similar decay rate for all spectra of $\xi \sim 30$ cm$^{-1}$, independent of the molecule, suggesting it originates from the metal nanostructure (Supplementary Figs. S5, S6). This is a surprising energy scale since it is much less than the thermal bandwidth ($k_B T = 202$ cm$^{-1}$). Possible origins include additional ERS components from $d$-band electrons in the Au, THz contributions to $\chi$ (less likely since $\hbar\xi \ll k_B T, E_F$), or from localised acoustic modes. The alternative that the additional component arises from Au-Au vibrations or from dielectric relaxational dynamics from liquid-like molecules is less convincing, since a distinct peak may be expected in such a case[37].

In previous work studying THz Raman spectra, the background was used to normalize the signal in a way that also facilitates the evaluation of effective local temperature[20], but here we are interested in removing all contributions except for the molecular Raman lines. After the above bosonic background removal, post-processing was used to select the stable representative 'nanocavity' spectra for each molecule from our large datasets. The Mahalanobis distance of each normalized spectrum to the centroid of the dataset for each molecule was calculated and used to select the thirty most representative nanocavity spectra (individual spectra shown in Supplementary Fig. S4)[38]. Comparing the THz region (25–200 cm$^{-1}$) of these thirty low-frequency spectra for each molecule, a variational Bayesian Gaussian Mixture model (see Supplementary Note 3)[39] infers seven clusters that correlate almost perfectly (normalized mutual information score 0.99) with their true molecular labels. This is evident when projecting into two dimensions using principal component analysis (PCA) for visualization (Fig. 2b), where each point is a spectrum with the centre coloured by its true molecular label and the edge coloured by its cluster label. Comparing the representative spectrum of each shows similar molecules are fully distinguishable even in the THz region both by inspection (Fig. 2a) and by machine-learning classification (Fig. 2b). This implies that the features are molecule-metal rather than nanostructure-dependent.

We observe that low-frequency peaks are significantly broader than those at higher frequencies (Fig. 3 and Supplementary Fig. S9), which suggests they are more sensitive to local conformation. Unpacking this rich behaviour offers improved ways to study intermolecular and molecule-metal interactions, and distinguish chemical enhancements, adsorption, surface reconstruction, and steric confinement. In order to develop this understanding, full calculations of the THz Raman spectrum are needed.

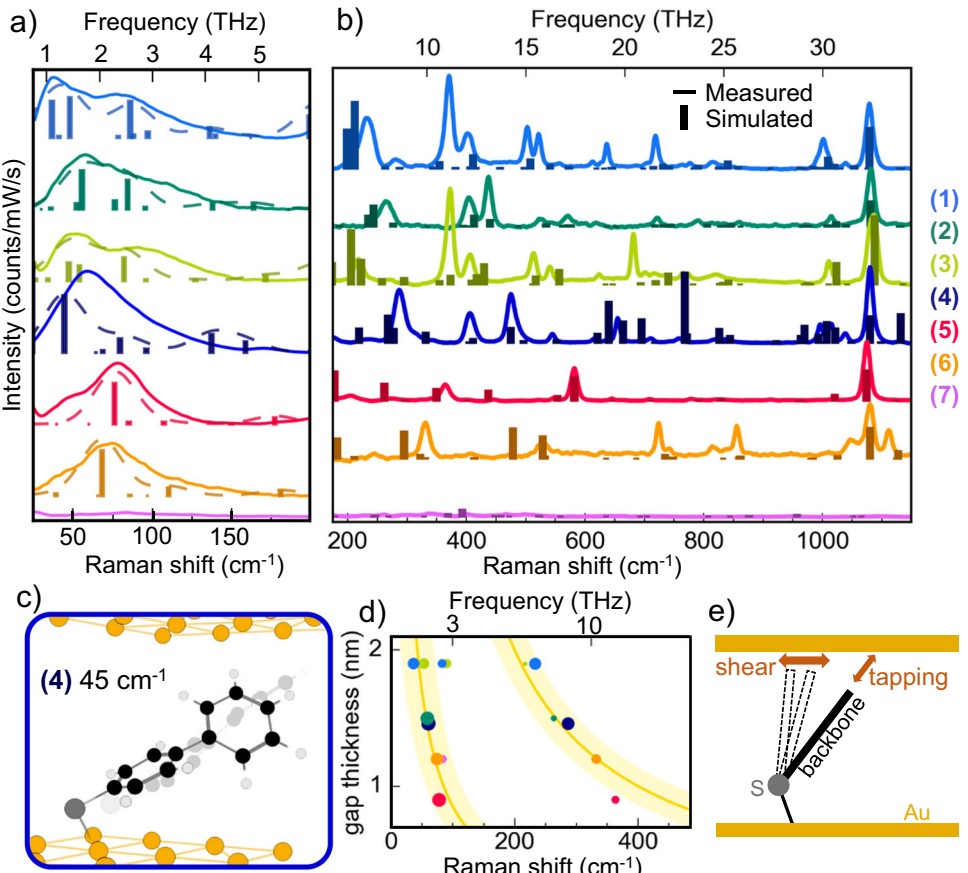

**Fig. 3 | Comparing THz vibrational modes with DFT simulation. a, b** Low- and high-frequency background-subtracted measured SERS spectra (lines) compared to simulated Raman peaks (dashed and bars, omitting modes < 25 cm⁻¹, dashed curves are Gaussian broadened by 32 cm⁻¹). An exponential function has been fit and subtracted in (**a**) as described in Supplementary Note 4 (from data in Fig. 2a). **c** Extremal positions of the atoms of an oscillating BPT molecule (**4**) for a THz

vibrational mode at 45 cm⁻¹ obtained with DFT (displacement amplified for visualisation). **d** Strongest measured THz modes compared to extracted gap thickness, as discussed in Supplementary Note 5 (colours as in (**a**)). Broad yellow lines show $1/L$ dependence from Eq.(2). **e** Vibrational motions are described by how molecules interact with the Au facet.

## Vibrational simulations

Molecular density functional theory (DFT) simulations at low frequencies are more challenging than above 500 cm⁻¹ since the motion of large blocks of atoms at those low frequencies are found to depend significantly on specific interactions with the metal facets. To properly account for the metal environment, we simulate a single molecule between two parallel blocks of Au atoms compatible with the Au (111) surface arrangement (the influence of the environment can be revealed by comparing these calculations with those of the molecule attached to a single Au atom or to a Au tetrahedral cluster in Supplementary Figs. S10, S11). The thiol headgroup is bound to a plane of fifteen gold atoms, and its tail group is allowed to relax close to a second parallel plane of sixteen gold atoms held a fixed distance from the first. The Au gap thickness for each molecule is varied and compared to the measured spectra (Supplementary Fig. S12), however, since the tilt angle is not fixed in DFT, the effects of gap thickness cannot be simply isolated. The excitation optical field is polarized perpendicular to each gold plane plane[22], and all simulated frequencies are scaled by a constant factor (0.986) to align the measured and simulated 1080 cm⁻¹ peak of BPT (molecule **4**) (Fig. 3). Molecule (**7**) is included here for contrast, as an example with negligible stable vibrational SERS peaks. The measured and simulated spectra are not shown here below 25 cm⁻¹ due to uncertainty in fitting the background in this region where Rayleigh scattering, VHG notch filters, and the vibrational and electronic Raman signals overlap.

These simulations are compared to the experiment after removing from the latter both bosonic and additional exponential backgrounds (Fig. 3a, b), as described above. In the THz regime (Fig. 3a), Gaussian-broadened simulated peaks (dashed) nicely agree with the broad measured features, though precise mode assignment is not always possible. The assignment is more straightforward at higher frequencies $\nu > 1000$ cm⁻¹ where the sharp lines are found to well match DFT (Fig. 3b, bars show DFT).

Based on our DFT simulations and in agreement with relevant literature[19,40], the strong peak observed near 1080 cm⁻¹ in the SERS of molecules (**1–6**) (Fig. 3b) is due to the longitudinal stretch of the C-S bond and adjacent phenyl ring (Ph). While the assignment of peaks below 1000 cm⁻¹ is less easy, strong peaks observed near 400 cm⁻¹ involve out-of-plane ring distortions in the single-molecule simulations, while those around 200–370 cm⁻¹ involve Ph stretches coupled to Au-S-Ph hinging, loaded by the various tailgroups (Fig. 3b). Our simulations of the seven aromatic thiols here each have five to ten vibrational modes below 200 cm⁻¹. The strongest of these involve simultaneously a rigid motion of the ring(s) and hinging around the Au-S-Ph bond (Fig. 3c and Supplementary Note 9).

These low-frequency modes cannot be described as simple bending or stretching of a particular bond, so we use an approximate model in order to discuss and visualize them. The simplest model for low-frequency modes of such molecules is a spring network of length $L$,[41] which is rigidly fixed at one end (Fig. 3e). In this case, the resulting single effective spring can support a series of harmonic vibrations,

described by

$$\nu_n = (v_s/L)(1+2n)/4 \qquad (2)$$

with acoustic wave velocity $v_s$ depending on average transverse and longitudinal spring constants, effective mass, and positive odd integer mode number $n$. Furthermore, the tail end of the molecule experiences either shear or tapping motions relative to the Au facet (Fig. 3e), involving very different interactions of the tail group with the Au. While in a previous work characterizing molecules of similar length, the effect of the molecular mass on $\nu_n$ was studied[41], here we emphasize the role of molecular length. The measured low-frequency peaks blue shift for shorter molecules ($L$) which give smaller gap sizes (Supplementary Note 5), as predicted by this effective spring-network approximation (Fig. 3d).

Divergence from this simple harmonic model is due to specific configuration complexities such as different Au-S-Ph angles, the three-dimensional motion of the rings, and interactions with the adjacent molecules. Typical THz vibrational motions of chain-like aromatic thiols from DFT (Fig. 3c) can be described as a combination of out-of-plane and in-plane ring distortions, mixed with torsional ring twists and hinging of the tilt angle. Ring in-plane stretching (resembling longitudinal spring-like vibrations) involves tapping motion of the tip atom relative to the top Au surface, and so is sensitive to the gap thickness and Au chemical interactions (Fig. 3e). Ring out-of-plane distortions (resembling transverse waves on a string) generally involve

a shearing motion of the tip atom relative to the top Au. In the spring-network picture, the top Au increasingly damps modes as the gap thickness decreases, because its proximity starts to constrain the tapping motion.

A general trend is that the modes become less localized to specific functional groups at low frequencies. This explains the increased broadening of measured low-$\nu$ spectral peaks. Delocalization means modes depend more on molecular conformation, metallic environment and molecule-molecule interactions, varying more from molecule to molecule in a SAM, and are harder to track using a single molecule picture. However, the results in Fig. 3a suggest that a full accounting of the gold planes surrounding single molecules in the theory captures well the main experimental features and identifies the main vibrational motions contributing to the spectra. In addition, we discuss below the effect of the interaction between molecules. The strong influence of the local gold configuration[19] in the Raman spectra is further emphasized in Supplementary Figs. S10–S12.

## From single-molecule towards collective vibrational modes

We address next the emergence of collective effects in the low-energy vibrational modes when the number of molecules considered in the cavity is increased, a situation that more appropriately mimics the SAM configuration. We carry out these challenging DFT simulations of model SAMs (Fig. 4) in the case of BPT (molecule **4**) (for a finite system of three molecules) and MBN (molecule **5**) (with four molecules) located between two parallel blocks of Au atoms arranged as the

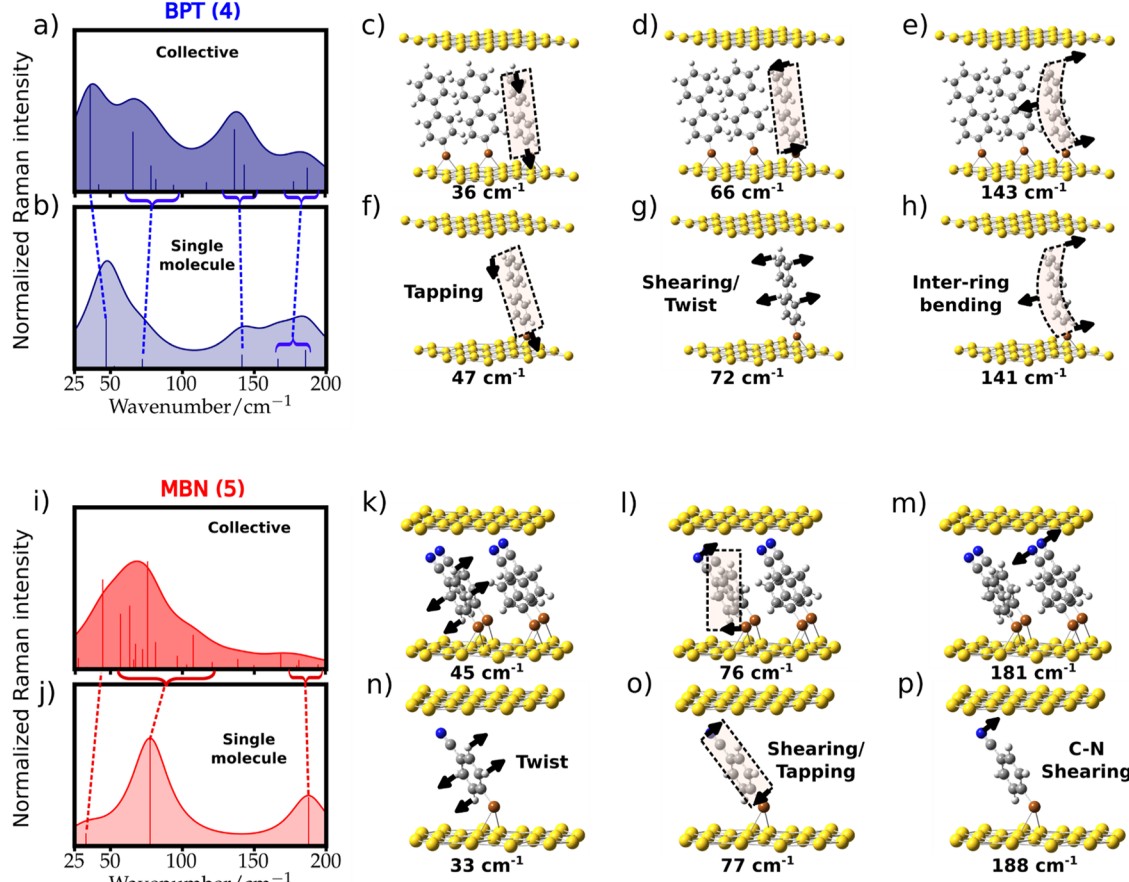

**Fig. 4 | Collective effects in Raman spectra.** Simulated Raman spectra for model SAMs based on (**a**) three BPT (molecule **4**) and (**i**) four MBN (molecule **5**), compared to the single molecules (**b, j**), when located between two flat gold layers. Simulated Raman peaks (vertical solid lines) are Lorentzian broadened by 32 cm⁻¹ for shaded curves, and dashed lines connect related vibrational modes of single molecule and model SAMs. (**c–h, k–p**) Vibrational motions of three dominant modes below 200 cm⁻¹ of single-molecule (**4: f–h**) and (**5: n–p**), with corresponding modes in model SAMs for molecule (**4: c–e**) and (**5: k–m**). Atoms coloured white (H), grey (C), blue (N), brown (S), and yellow (Au), black arrows point in the direction of atomic displacements, and pale boxes denote deformations of the entire molecule.

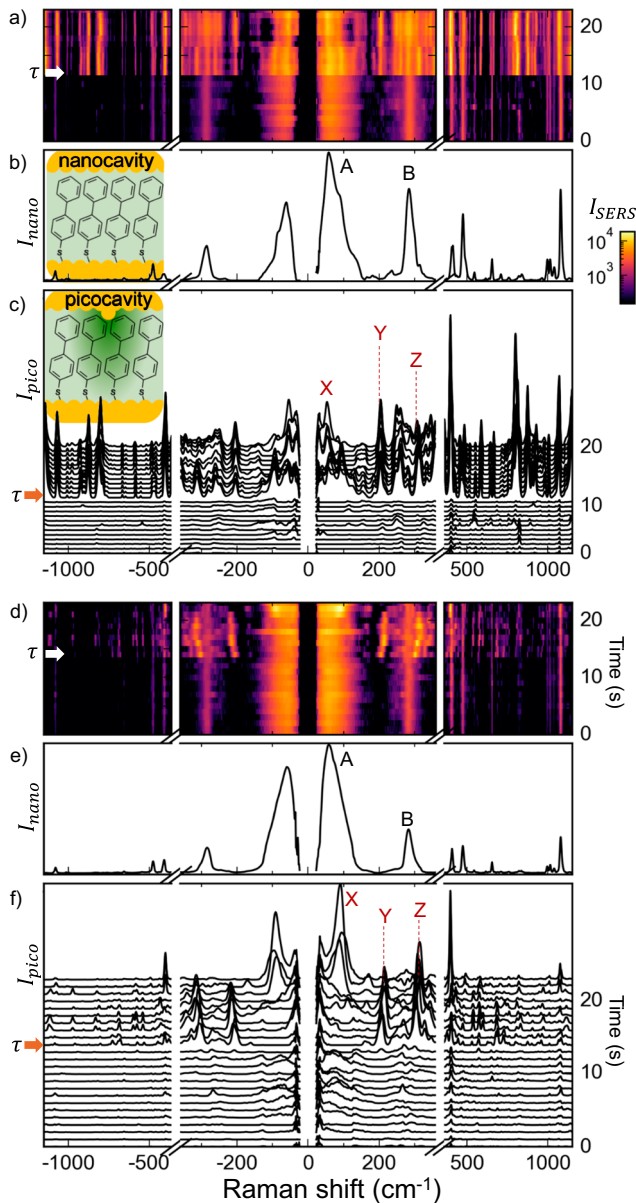

**Fig. 5 | Picocavity THZ SERS.** showing two examples (in **a**–**c**, **d**–**f**) using BPT (see inset, molecule 4). **a, d** Time series of consecutive SERS spectra (1 s integration time) with picocavity induced at time $\tau$, giving new single molecule SERS peaks. **b, e** Extracted nanocavity SERS spectrum. **c, f** Picocavity SERS (extracted by subtracting from each spectra in (**a, d**) the average nanocavity spectrum, taken from times 0–10 s). Inset in (**c**): Atomic protrusion from the top NP facet forms picocavity with enhanced field (darker green) compared to the nanocavity. The bosonic and exponential background contributions are subtracted in all spectra.

Au(111) surface. Intermolecular interactions split the THz vibrations of the single molecule (increasing the number of vibrational modes), and modify the relative Raman intensities. The displacements in these split modes are related to the single-molecule motion, but often with more complex hybridized patterns in the collective modes (see Supplementary Note 8 for full description). The intermolecular coupling depends on both the molecular structure and the environment and produces a redistribution of the modes and, thus an overall broadening of the Raman spectra.

In the case of BPT (molecule **4**), the Raman spectrum for three molecules in the cavity (Fig. 4a) gives a set of peaks originating from split modes with respect to those of the original single molecule (see

blue dashed lines connecting related vibrational modes between Fig. 4a, b). The single-molecule tapping (Fig. 4f), shearing (Fig. 4g), and inter-ring bending (Fig. 4h) modes at $47\,\text{cm}^{-1}$, $72\,\text{cm}^{-1}$, and $141\,\text{cm}^{-1}$ respectively, give rise to groups of modes around these energies with similar vibrational displacements as their single molecule constituents, but with in-phase and antiphase relative motions between the molecules (see Fig. 4c–e, and Supplementary Note 8 of these collective modes). Similarly, in the case of MBN (molecule **5**), we observe that the collective spectrum for four molecules (Fig. 4i) shows broader peaks than the single-molecule reference (Fig. 4j), which arise from multiple modes related to the original single-molecule motions (see red dashed lines connecting these). These resemblances are seen from the displacements in the three main THz modes of the single molecule (**5**) (Fig. 4n–p), together with representative modes for each in the collective case (Fig. 4k–m). The examples above indicate that intermolecular interactions lead to collective vibrational excitations in the SAMs that can be traced back to the single-molecule Raman spectra, even though they produce a broader and richer structure of Raman lines.

We emphasise the demands of modelling in this THz regime, which challenge the capability of tractable simulations. The Au facets must be included as they have a significant role due to covalent and Coulombic interactions. Investigating the effect of gap thickness with only a single molecule (Supplementary Fig. S12) further indicates the dependence of the Raman spectra on the details of the molecular surroundings but induces unrealistic bending at the S atom. Modelling intermolecular interactions, which are particularly important in the THz region requires that periodic unit cells contain more than one molecule, which greatly extends simulation times. This is in contrast to higher frequencies where DFT gives intermolecular shifts[40] of $<10\,\text{cm}^{-1}$, so single molecule simulations suffice. Other influences may include mode anharmonicity, presence of water, and ions. Finding the system ground state is also an issue for smaller unit cells, given the extensive debate about the reconstruction of the Au (111) surface.

### Single molecule THz vibrations

In order to suppress the broadening in this THz range induced by the local environment (from extended metal-molecule interactions as well as inter-molecular collective effects), we measure the SERS of individual molecules (one or very few) within the SAM. To do this, additional atomic-scale confinement of light is utilised, formed around adatoms on the Au surface that can be optically induced in these structures, as shown by many recent papers and summarised in ref. 42. Such picoscale optical cavities (termed 'picocavities') produce a dramatic hundred-fold increase in intensity from selected Raman lines and anomalously large anti-Stokes-to-Stokes intensity ratios[43]. The picocavity-induced field enhancement is confined to regions with a size of approximately an atom; nearby molecules outside this region may affect the THz Raman spectra due to the collective nature of the low-energy vibrational modes, but the SERS signal should still be dominated by a very small number of molecules. Indeed, we show below that picocavity SERS lines do not suffer line broadening due to averaging over a heterogeneous distribution, as expected from this discussion. These SERS lines do, however, exhibit variable frequency shifts due to the different possible picocavity configurations and relative orientations of adatom and molecule, which can vary over time.

The linewidths of typical SERS peaks above $200\,\text{cm}^{-1}$ are $<20\,\text{cm}^{-1}$. To better identify these new picocavity lines also in the THz spectral range, we first extract the nanocavity spectrum by excluding transients in the time series SERS data and identify the spectrum that is stable in time. Subtracting this nanocavity spectrum produced by the few hundred molecules within the nanogap (Fig. 5b, e) from the total SERS (Fig. 5a, d) distinguishes the additional SERS peaks (Fig. 5c, f) associated with the picocavities. While the linewidths of the nanocavity

peaks increase threefold below 200 cm$^{-1}$, the picocavity linewidths remain the same across the entire spectrum (Supplementary Fig. S9). This confirms that the intrinsic linewidth of single-molecule vibrations in the THz region is consistent with that in the infrared, and the broad peaks come from differences in the local environment of each molecule, differences in molecular conformation, or from mode splitting induced by intermolecular interactions (rather than for instance significant mode anharmonicity). From a different perspective, the lack of inhomogeneous broadening firmly supports the expectation that the picocavity is probing the THz modes of very few molecules. So far, single molecule vibrations at low frequencies have not been investigated, to our knowledge, and our work suggests that even in confined nanogaps, vibrational decay is not dramatically faster in the THz region. Previous work indicates that large anharmonicities can be found for low frequency (such as torsional) modes[44], which increase mode mixing and thus decay rates, but since the most extensive work using optical Kerr effect pump-probe measurements[45] is on large ensembles, the role of heterogenous environments is not yet known.

In the particular THz SERS dynamic tracks of molecule (**4**) shown in Fig. 5, new spectrally narrow lines X, Y, and Z (which are related to stable nanocavity lines A, B) appear after 10 s due to picocavity formation, along with many other lines at higher frequencies. Variations in the vibrational frequency of the new lines have been previously shown to relate to the transient movement of the Au adatom relative to the molecule, perturbing the metal-organic coordination bond and, thereby, the electronic structure of the molecule[39]. We highlight that these narrow lines (much narrower than the nanocavity line A) emerge even below 100 cm$^{-1}$ (line X). As noted above, a complete interpretation of these dynamics requires a full accounting of the molecular assembly and configuration of the atomic protrusion, but this data clearly shows the experimental capability to track metal-molecule interactions in real-time.

In conclusion, we use THz Raman spectroscopy on reproducible plasmonic nanocavities containing monolayers of different aromatic thiols to investigate their low-frequency vibrational modes. While low-frequency Raman spectroscopy has been widely employed in studying two-dimensional materials and molecular spectroscopy, we highlight our robust experimental approach and data processing (obtaining clear low-frequency Raman signals) when combined with plasmonic near-field enhancement and localization. We show that the background SERS signal induced by the strong localised plasmonic fields inside the metal follows the frequency dependence characteristic of a bosonic distribution, providing evidence for contributions to the spectra from metal electronic Raman scattering, as well as uncovering an additional component with exponential dependence at the energy scale of 4 meV. The low-frequency molecular signatures are shown to be consistent and fully distinguishable, allowing their spectral shapes to be meaningfully compared to DFT. These calculations indicate that THz Raman peaks are very sensitive to the molecular environment, pointing out the influence of intermolecular collective interactions and the importance of capturing in simulations the appropriate metal-molecule configuration at the gold surfaces. Our calculations also serve to classify the THz vibrations and emphasize two types of motion relative to the top Au surface, producing shear- (scraping the tip atom across the facet) and tapping motions (resembling tapping-mode AFM). A picture thus emerges of the molecule as a network of springs, with distinctive behaviours of the complicated delocalized THz modes. We show that these motions are somehow preserved even with intermolecular interactions. Single- (or few-) molecule measurements of THz SERS enabled by the formation of picocavities show that narrow intrinsic lines lie inside the environmentally broadened low-frequency spectrum, giving a suitable metric for molecular order in such SAMs. Since charge transport through such molecules can modify vibrational energies, this work opens up the possibility to study molecular conformation changes coupled to redox processes.

## Methods

### Sample preparation
The flat Au(111) surface is prepared by a template-stripping method. Approximately 100 nm of Au is evaporated at a rate of 1.0 Å/s onto a Si wafer (Si-Mat) with typical roughness < 3 Å. Pre-cut 10 × 5 x 0.5- mm glass slides (UQG Optics) are then attached to this surface with UV glue (Norland 81) and cured with UV light (364 nm) for 35 min. The glass pieces can then be removed by cutting the perimeter with a blade, exposing a smooth Au surface with rms roughness < 0.2 nm.

Molecules (**1**) through (**6**) (Sigma-Aldrich) and molecule (**7**) (Enamine) were used as purchased. A SAM is formed on the Au surface by submerging the Au-coated glass into a 1 mM solution of the molecular crystal powder dissolved in anhydrous ethyl alcohol (Sigma-Aldrich > 99.5%) for 16–24 h, with the exception of (**2**), which requires immersion in a 0.5 mM solution for 1 h. They are then rinsed with the same solvent and dried in $N_2$. Colloidal 80 nm diameter nanoparticles (BBI Solutions) are drop cast onto the SAM-covered surface, then rinsed after 30 s with deionized water, and $N_2$ dried. If sparse deposition occurs, the final step is repeated with a ratio of 10:60 μL $NaNO_3$ to nanoparticle solution.

### Spectroscopic measurements
A custom system was built to measure SERS down to 5 cm$^{-1}$ (Fig. 1a). A 785- nm diode laser (Integrated Optics, Matchbox Model 785L-21A) with external thermoelectric cooler (Integrated Optics, AM-H9) is collimated before passing through a pair of volume holographic grating (VHG) clean-up filters (OptiGrate BPF-785, FWHM < 0.12 nm) mounted on manual rotation stages with micrometres (Thorlabs) for angle tuning. Two gratings are required for blocking all laser light outside the main laser line. In the configuration shown in Fig.1a, they are angle-tuned in a counter-rotating fashion to avoid beam displacement into the microscope. The beam is coupled into a microscope (Olympus BX53M) fitted with a high-NA objective (Olympus LMPFLN100xBD, 0.8 NA) suitable for both SERS and darkfield spectroscopy. Maximum CW laser powers in the sub-μm diameter focal spot are 50 μW. A halogen lamp is coupled through a darkfield mirror cube into the objective to measure the NPoM scattering spectrum for sample characterization. A x-y motorized stage (Prior Model PS3J100/ D) enables automated measurements of many hundred single particles. Backscattered light is collected by the objective and passed through a series of three VHG notch filters (OptiGrate BNF-785, FWHM < 0.6 nm, OD3) separated by irises to minimize the collection of spurious scattering. These three notch filters are required to reject the narrow laser line, which is at least six orders of magnitude more intense than the Raman scattering while transmitting Raman scattering down to ± 5 cm$^{-1}$ to a single-grating spectrometer (Horiba Triax 550, 600 l/mm grating) coupled to a 2048 × 512-pixel front-illuminated CCD (Andor Newton Model DU940P-FI).

### Spectral pre-processing
The measured inelastically scattered light ($I$) contains additive contributions of Raman scattering from the molecules in the SAM in the NPoM ($I_M$) multiplied by an enhancement factor ($EF$) given by the optical field enhancement in the nanogap, and other sources of inelastic scattered light ($I_{background}$),

$$I = (EF) . I_M + I_{background} \qquad (3)$$

Subtraction of the smooth background obtained by polynomial fitting or other computational methods does not necessarily result in the true spectrum of interest. As discussed, we posit that the background is emitted primarily by surface-enhanced ERS [Eq. (1)] with an additional weak exponential component. Our correction algorithm to remove the background relies on physical constraints to make initial guesses for and set bounds on the parameters. $T$ is constrained to temperatures

between 288 and 340 K, and the weak broad background Gaussian centre and width are set to those of the plasmon resonance measured for each NPoM using DF spectroscopy. A customized simulated-annealing random search method is used to fit the model, with an infinite penalty applied to fit that cross above the spectral background.

## DFT Simulations

To simulate the Raman spectra of the seven aromatic thiol molecules in vacuum (without gold atoms bound to the thiols) and the SERS spectra of the same molecules attached to Au atoms, we carry out density functional theory (DFT) calculations in two steps, using the code Gaussian16 Rev B.01[46]: we first optimise the structure of the molecules in vacuum and with gold structures attached to their thiols; we then compute the vibrational frequencies and the Raman polarizability tensors for the optimised structures.

We optimise the molecular structure in vacuum using the DFT standard B3LYP exchange-correlation-functional[47] and the basis set 6–31 G(d,p) for the atoms of hydrogen, carbon, nitrogen, oxygen, fluorine, sulphur, and bromine. To simulate the SERS spectra of the molecules interacting with Au, we first optimise the structure of the selected molecules bound to different gold structures. The structures simulated here include two parallel monolayer planes of fifteen and sixteen gold atoms, two tetrahedral clusters of twenty and nineteen gold atoms with the molecule bound to the inner apex, and single gold atoms (see Supplementary Fig. S10 for sketches of the structures). To obtain the relaxed structure of the monolayer planes, we first carry out calculations under periodic boundary conditions for molecule 4 in a cavity formed by two gold (111) surfaces with the code VASP (version 5.4.4)[48–50]. In order to obtain this structure, we use a unit cell formed by 8 layers of a $2 \times 3$ Au(111) surface to describe each of the two slabs that form the cavity. For periodic calculations, we use plane waves as a basis set, with a cutoff energy of 420 eV and the PAW pseudopotentials from the VASP database[51,52]. We sample the reciprocal space using $2 \times 3$ points following the Monkhorst-Pack scheme. As a function, we choose OPTPBE[53] in order to take into account weak interactions (van der Waals forces). We impose an electronic convergence of $10^{-6}$ eV and, for the forces, a criterion of 0.01 eV/Å for all the degrees of freedom of the molecule and the $z$-direction perpendicular to the layers that delimit the cavity. We then optimise the structure of the molecules sandwiched between the gold structures using the same exchange-correlation functional and basis set that we employ to optimise the structure of the molecules in vacuum within the code Gaussian16. For the atoms of gold, we use the LANL2DZ basis set[54]. To account for the non-covalent interactions between the molecules and the gold structures, we add the empirical dispersion correction D3 of Grimme[55] with the damping function of Becke and Johnson[56,57]. To avoid distortions of the gold structures during the optimisation that lead to unphysical conformations, we freeze the position of the gold atoms.

The simulations of the Raman signal of BPT and MBN (molecules 4, 5) mimicking SAMs use 3 or 4 molecules, respectively. We obtain the metal-molecule configuration of the structure holding these molecules following the same procedure used above for infinite (periodic) monolayers, except we do not perform the last optimization with Gaussian16. We also proceed in this way for the single-molecule calculations (same molecules 4, 5) used as reference in Fig. 4, and in the corresponding Supporting animations.

Once the molecular structure has been obtained, we compute the Stokes Raman cross-section assuming that the Raman dipole of the system formed by the molecule and the gold structures emit in free-space,

$$\frac{d\sigma_k^{Raman}}{d\Omega} = \frac{\hbar(\omega_{inc} - \omega_k^{Raman})^4}{16c_0^4\varepsilon_0^2} \frac{\left|\vec{e}^{rad}\overset{\leftrightarrow}{\alpha}_k^{Raman}\vec{e}^{inc}\right|^2}{\left(1 - e^{-\hbar\omega_k^{Raman}/k_BT}\right)} \quad (4)$$

with $\varepsilon_0$ the vacuum permittivity, $c_0$ the speed of light in free space, $k_B$ the Boltzmann constant, $\hbar$ the reduced Planck constant, $T$ the temperature, $\omega_{inc}$ the angular frequency of the incident electromagnetic field and $\omega_k^{Raman}$ the angular frequency of the vibrational mode $k$. $\overset{\leftrightarrow}{\alpha}_k^{Raman}$ is the Raman polarizability tensor of the vibrational mode $k$, and $\vec{e}^{inc}$ and $\vec{e}^{rad}$ are unit vectors along the direction of the polarization of the incident and scattered electric fields, respectively.

To calculate the Stokes Raman cross-section, we assume a temperature $T = 298.5$ K, incoming light of angular frequency $\omega_{inc} = 2399 \times 10^{12}$ rad s$^{-1}$ (785 nm), and the values of $\overset{\leftrightarrow}{\alpha}_k^{Raman}$ and $\omega_k^{Raman}$ are obtained from DFT calculations with the exchange-correlation-functional B3LYP and the basis set 6–31 G (d, p) as above. For the atoms of gold, we use the basis set LANL2DZ. To isolate the native molecular vibrational modes from those modes that couple the vibrations of the molecule and the gold structures, we also freeze the position of the gold atoms in these calculations. We consider that the polarization of the scattered light is parallel to the polarization of the incident light ($\vec{e}^{inc} = \vec{e}^{rad}$). We use different polarization directions depending on the gold structure bound to the molecule: (i) perpendicular to the gold layers (for gold planes), (ii) parallel to the line that connects the inner apexes (for tetrahedral gold clusters), and (iii) parallel to the axis that connects the thiol headgroup to the tail group for the structure formed by a single gold atom (see Supplementary Fig. S10). We also use the latter polarization direction to compute the Stokes-Raman cross-section for the molecules in vacuum (no gold atoms).

To obtain the Raman spectra for the model SAMs and single-molecule for BPT (molecule 4) and MBN (molecule 5) in Fig. 4, we remove the Raman lines at wavenumbers below 25 cm$^{-1}$ and normalize to the maxima in the range 25–200 cm$^{-1}$. We then scale the simulated frequencies by a constant factor to align the measured and simulated peaks at 1080 cm$^{-1}$, which corresponds to the C-S stretching mode. We use the scaling factors 0.992, 0.992, 0.993 and 0.992 for the results in Fig. 4a, b, i, and j, respectively.

All three-dimensional representations of molecules are produced using ChemCraft (www.chemcraftprog.com), while two-dimensional ones use ChemDraw (PerkinElmer Informatics).

## Data availability

The data that support the findings of this study are available from the corresponding author and the data is deposited in the Cambridge Open Data archive[58].

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

## Acknowledgements

Funding: J.J.B. acknowledges support from the European Research Council (ERC) under Horizon 2020 research and innovation programme PICOFORCE (883703) and POSEIDON (861950) and from the UK EPSRC (Cambridge NanoDTC EP/L015978/1, EP/L027151/1, EP/S022953/1, EP/P029426/1, and EP/R020965/1. R.A.B., R.E. and J.A. acknowledge support from PID2019-107432 GB-I00 and PID2022-139579NB-I00 funded by MCIN/AEI /10.13039/501100011033/ and by "ERDF A way of making Europe" grant IT 1526-22 from the Basque Government for consolidated groups of the Basque University, and Elkartek project u4Smart of the Department of Economy Development of the Basque Country. BdN acknowledges support from the Leverhulme Trust and Isaac Newton Trust in the form of an ECF. F.A.G. acknowledges funding from PID2022-138470NB-I00 funded by MCIN/AEI, and computing time from Red Española de Supercomputacion through the project QHS-2021-2-0019 and Universidad Autónoma de Madrid for funding a research stay with the "Recualificación" programme (CA5/RSUE/2022-00234).

## Author contributions

All authors contributed to the design and realization of this project. Experimental measurements were taken by A.B.A., designed by A.B.A., B.d.N. and J.J.B., and experimental automation by E.E. Data analysis was performed by ALB and J.J.B. DFT simulations were performed and analysed by R.A.B., F.A.G., R.E., T.F., E.R. and J.A. All authors contributed to writing and editing the manuscript.

## Competing interests

The authors declare no competing interests.
