## [Peer Review File · Nature Communications]

Uncovering low-frequency vibrations in surface-enhanced Raman of organic moleculesREVIEWER COMMENTS

Reviewer #1 (Remarks to the Author):

This manuscript reports on SERS observation of terahertz vibrations of aromatic thiol monolayers on Au surface. The authors successfully obtained such low-frequency vibration spectra by subtracting the backgrounds, which were generated by the fitting with the expression for ERS. The observed low-frequency modes were identified as shear and tapping modes by assistance of DFT simulations.

However, very similar results have already been reported by Ikeda et al. For the origin of the background continuum in SERS spectra, they have reported that the asymmetric background can be well fitted in the wide frequency range including THz region by the ERS model (Chem. Sci. 11, 9808(2020)/Phys. Status Solidi B, 259, 2100589 (2022)). Based on this, they have also reported that low-frequency vibrations can be uncovered by reducing the Bose-Einstein thermal factor from a measured spectrum in these papers. An important point is that the ERS background is not subtracted but divided by the BE thermal factor in the Ikeda's method. It should be noted that any baseline subtraction may lead to spurious bands appearing or bands disappearing from a measured spectrum. In this sense, the previously reported method is better than the present method. Moreover, they have reported simultaneous observation of VRS from molecular monolayers and ERS from Au under electrochemical conditions using this method (Nanoscale 12, 22988 (2020)). In addition, the shear mode of aromatic thiols has been thoroughly studied in Catal. Sci. & Technol. 12, 2670 (2022) although their mode description is slightly different.

In conclusion, the reviewer cannot find any novelty required for publication in this journal.

Reviewer #2 (Remarks to the Author):

The manuscript titled "Uncovering low-frequency vibrations in surface-enhanced Raman of organic molecules" reports a study on elucidating terahertz (THz) range molecular vibrational spectra in low frequency surface enhanced Raman spectroscopy (SERS), combining optical measurements and density functional theory calculations. Expanding the spectra access of SERS into THz is a new frontier for an interdisciplinary research community. The spectral behaviors of SERS in the THz exhibit certain qualitatively different features than those observed in the conventional frequency range. Thus, a thorough fundamental understanding is crucial for the field to move forward and for low frequency SERS to find broad applications. This manuscript reports interesting findings from thoughtfully designed experimental systems and provides good insight in understanding this new spectroscopic technique. With these noted, there are a number of issues that prevent a favorable recommendation, as discussed in detail below.

1. In the introductory text, the authors stated "...we push it to its low frequency limit, which is rarely utilized, ..." Some of the references of this manuscript overlap with the current study in their scope, namely low frequency / THz SERS – for example, Inagaki et al, Curr Opin Electrochem, 2019 (ref 12), Inagaki et al JPCL 2017 (ref22), Sun et al JPCC 2022 (ref23). It would be better if the authors acknowledge these prior works and use them to better provide context to the current work.
2. A strength of this study is the automated spectral acquisition from hundreds of particles. This allows more robust measurements and comprehensive sampling of molecular arrangements. In the manuscript and SI, mainly individual representative spectra were shown. One might view the selection of representative spectra as a form of ensemble averaging. Yet the strength of single particle measurements is to reveal individual variations, which are largely absent in the current presentation. This reviewer encourages the authors to also inform the reader about the variation and distribution of the spectra in the whole dataset.

3. In equation 1, the authors included both the Bose-Einstein (BE) and Fermi-Dirac (FD) distribution in modeling the spectral background. The BE distribution carries a T dependence which is used to infer the electronic temperature. The FD distribution, however, is represented using just a step function with no temperature dependence. Why?

4. To analyze the low frequency SERS spectra, the authors adopted an unsupervised clustering algorithm implemented in the standard machine learning python library. Were full spectra used as input vectors to the clustering algorithm?

It is clear the high frequency portions of the SERS spectra contain local vibrations from the thiol molecules. It would be interesting and informative to segment each SERS spectra into high frequency and low frequency portions and repeat the clustering analysis. Cross comparing the three cases (whole / high frequency / low frequency) would allow one to discuss the nature of spectral features more comprehensively. It would enable the authors to more clearly separate the spectral components in the low frequency range between molecular contributions and plasmonic metal contributions. Would one see both molecule-independent and molecule-dependent features?

5. What is the meaning of n in the spring network model? It was not defined in the text.

6. In the low frequency range, one could expect to observe two types of vibrations. One is the collective and global motion of individual molecules, the other is the external or lattice modes of molecules vibrating in an ordered lattice. The self-assembled monolayers (SAMs) of thiol molecules are known to exhibit good packing with a periodic lattice.

One might argue that the DFT calculation should be done using a relaxed crystalline lattice. In this work VASP is used only to obtain the relaxed structure, the spectral calculation is done using Gaussian, on isolated single molecular constructs. What is the justification for this choice? Wouldn't this choice of computation method entirely miss the lattice modes from the packed thiol molecules? There seems to be low frequency peaks, unaccounted for in the molecular calculation.

7. Figure 3 a and b consistency. Figure 3a shows the measured spectra overlaid with Gaussian broadened DFT predictions. Figure 3b shows the measured spectra with a stick plot of simulated spectra. Please also show the stick plot predicted in the low frequency range.

8. Template stripping. The flat Au surface obtained from template stripping is noted to be (111). Could the authors provide literature reference or data to support the identification of the Au film's orientation?

In summary, this is an interesting report on the development of basic understanding of SERS in the THz frequency range. However, the issues listed above undermine the quality of the scientific discussion and dampen the enthusiasm. This reviewer recommends reconsideration after major revisions.

Reviewer #3 (Remarks to the Author):

Comments attached.

Comments on “Uncovering low-frequency vibrations in surface-enhanced Raman of organic molecules”

In this work, the authors employ THz SERS spectroscopy to investigate the low-frequency vibrational modes of molecular monolayers confined inside single plasmonic nano- and picocavities. The THz vibrations of the confined molecules, which can be assigned using density functional theory (DFT) calculations, are fully distinguishable and consistent with that in the infrared. In particular, the pico-cavity SERS spectra exhibit narrower peaks in the low-frequency region than that in the nanocavity, revealing the vital role of local environment in the molecular conformation. Overall, the experiment is well designed with adequate theoretical analysis using appropriate models. Therefore, I would like to recommend this work for publication in Nature Communications after the authors address the following comments properly.

1. The graphs in the article would benefit from further optimization. For instance, the experimental setup diagram does not provide a clear illustration on the role of some components in the developed THz SERS spectroscopy, such as the rationale behind using two volume holographic gratings (VHG) clean-up filters after the laser beam and three VHG notch filters in the collection path. Additionally, there is an inconsistency in the schematics above Fig. 1b and Fig. 1d, which needs further clarification. As shown in Fig. 1c, ν represents vibrational frequency, which is in general a positive number. Accordingly, Stokes and anti-Stokes Raman shifts should be expressed as $\omega_L - \nu$ and $\omega_L + \nu$, respectively. However, both the schematic above Fig. 1b and $\nu < 0$ assumed in Eq. 1 are different from such conventions. It is recommended that the author adjust the schematic diagram of Fig. 1b according to the generally accepted representation method to make it consistent with that of Fig. 1d.
2. On page 3, the authors claim that THz vibrational modes strongly depend on molecular conformation - why is it the case particularly in THz region? The authors attribute the large FWHM of low-frequency Raman peaks to their local molecular conformation. Please provide further discussions to better illustrate this argument.
3. In Fig. 2, the low-frequency SERS spectra of seven aromatic thiol molecules inside gold NPoMs were measured and compared. Different lengths and functional groups of these molecules result in varied effective compressibility. Please explain the physical meaning of “compressibility”. In addition, the different junction conductance in these molecule-sandwiched gold NPoMs could also affect the measured SERS spectra, potentially due to electron transport through such ultrasmall plasmonic molecular nanogaps under light illumination. Please refer to ACS nano, 2018, 12(7): 6492-6503 for detailed information and make relevant discussions.
4. Regarding Figs. 2a and 3a, it seems that the background-subtracted low-frequency modes in Fig. 3a are quite different from those original spectra in Fig. 2a. Please confirm whether the original data used for background subtraction in fig. 3a are the same as that in Fig. 2a; if not, please explain why different sets of data are used for comparison. Otherwise, the comparison with the simulated Gaussian-broadened Raman peaks would be meaningless. Please also add a paragraph in SI to illustrate the detailed procedures of background

subtraction.

5. The authors mention that submerging the Au-coated glass in the molecular solution for 16-24 h or 1 h results in the adsorption of SAM on the surface of gold. In Table S3, the gap thicknesses of gold NPoMs set by each SAM is calculated based on the measured dark-field scattering spectra and an NPoM quasi-normal mode tool. It would be better to provide detailed explanations on the calculation process. Besides, in Fig. 3d, the blue shifts in the THz vibration modes as a function of the extracted gap thicknesses is explained by a spring network model. Please explain this model and its validity range and limitation.
6. In Fig. 3a and Fig. 4, the authors fail to provide a clear explanation on the assignments of measured SERS peaks for the seven molecules in the THz region. Please add the assignments of molecular conformation or vibrational modes of the observed THz SERS peaks to better explore the origin of pico-cavity-induced low-frequency vibrations in the molecule-sandwiched NPoMs.
7. Figure 4 demonstrates that the molecular configuration can be effectively characterized by the low-frequency Raman modes. Interestingly, not only do the peaks below 200 cm^{-1} change with molecular vibrations, the peaks above 200 cm^{-1} also exhibit changes that can be utilized for configuration characterization. So what are the distinct advantages of utilizing the low-frequency vibrational signals for such characterization purposes? In addition, it is unclear from Figs. 4d-4f which molecule is used to form the molecular junctions. Finally, it is better to demonstrate how the nanocavity and pico-cavity SERS spectra were extracted from the time series of SERS spectra.
8. The THz SERS spectra shown in Fig. 4 change significantly after the laser irradiation for 10 s. Is it solely due to the transformation from nanocavity to pico-cavity (enabling single-molecule SERS) or also caused by potential damage of molecular junctions? Additionally, the linewidths of low-frequency SERS peaks in the pico-cavity are much narrower than that in the nanocavity - the authors should make further discussions on this phenomenon in the context of (single-molecule) SERS.
9. The authors present a robust experimental approach and data processing method for obtaining clear Raman signals from molecule-sandwiched NPoM structures at low frequencies. While low-frequency Raman spectroscopy has been widely employed in the field of two-dimensional materials and molecular spectroscopy studies, it is important to highlight the unique contribution of this method and its significance when combined with plasmonic near-field enhancement and localization.
10. The authors discuss in the main text that the low-frequency vibrational modes typically correspond to the complicated delocalized modes of molecular monolayers. However, the DFT simulation results shown in Fig. 3 are for single molecules, which may not accurately reflect the experimental results associated with molecular monolayers where multiple molecules are present in the cavities. Additionally, the spring network model seems to struggle to explain the findings of the pico-cavity with a molecular monolayer shown in Fig. 4c.
11. The present study utilizes plasmonic pico-cavities of ultrasmall mode volumes to probe molecule conformation through measuring low-frequency vibration modes below 200 cm^{-1} . In a related study published in *J. Phys. Chem. Lett.* 2019, 10(16), 4692–4698, it was demonstrated that single plasmonic nanoparticles, such as gold nanorods with a much larger

mode volume, could also serve as a SERS probe of molecular conformation and molecule-metal bonding configurations. Therefore, it would be interesting to compare this earlier study with the current one and elucidate specific advantages and necessity of employing plasmonic nanocavities and pico-cavities for similar purpose.

12. It has recently been demonstrated that the light-triggered conformation change of photoactive molecules confined inside a plasmonic nanocavity lead to reversible tuning of both SERS and SHG signals, <https://doi.org/10.1021/acs.nanolett.2c04988>. The author may also refer to this work and discuss how these low-frequency vibrational modes observed in the present nanocavities and pico-cavities affect their nonlinear emissions.
13. In the first paragraph of page 6, some references should be added to support the assignments of Raman peaks.
14. Please review and carefully check the citations in both manuscript and supporting information. For example, references 4 and 5 are identical in supporting information, and “Table S2” on page 3 of the text needs to be corrected to “Table S3”.
15. There are some mistakes in the text. Equations should be written in proper format, $n_{BE} = [\exp\{h\nu/k_B T\} - 1]^{-1}$ should be $n_{BE} = [\exp(h\nu/k_B T) - 1]^{-1}$.

Response to referees:

We greatly appreciate the feedback from all reviewers who emphasise that the work contributes to *“a new frontier for an interdisciplinary research community”* and is *“crucial for the field to move forward and for low frequency SERS to find broad applications,”* that this manuscript *“reports interesting findings from thoughtfully designed experimental systems and provides good insight in understanding this new spectroscopic technique”* (reviewer #2), *“the experiment is well designed”* and therefore *“recommend this work for publication in Nature Communications”* (reviewer #3). We answer all points in detail below.

Reviewer #1:

1. The authors successfully obtained such low-frequency vibration spectra by subtracting the backgrounds, which were generated by the fitting with the expression for ERS. The observed low-frequency modes were identified as shear and tapping modes by assistance of DFT simulations. However, very similar results have already been reported by Ikeda et al. For the origin of the background continuum in SERS spectra, they have reported that the asymmetric background can be well fitted in the wide frequency range including THz region by the ERS model (Chem. Sci. 11, 9808(2020)/Phys. Status Solidi B, 259, 2100589 (2022)). Based on this, they have also reported that low-frequency vibrations can be uncovered by reducing the Bose-Einstein thermal factor from a measured spectrum in these papers. An important point is that the ERS background is not subtracted but divided by the BE thermal factor in the Ikeda's method. It should be noted that any baseline subtraction may lead to spurious bands appearing or bands disappearing from a measured spectrum. In this sense, the previously reported method is better than the present method. Moreover, they have reported simultaneous observation of VRS from molecular monolayers and ERS from Au under electrochemical conditions using this method (Nanoscale 12, 22988 (2020)). In addition, the shear mode of aromatic thiols has been thoroughly studied in Catal. Sci. & Technol. 12, 2670 (2022) although their mode description is slightly different. In conclusion, the reviewer cannot find any novelty required for publication in this journal.

> We indeed appreciate the connection to papers by Ikeda et al (cited already as [12] and [13]), and now include also Catal.Sci.&Tech.12, 2670 (2022), but these however combine all modes below 100cm^{-1} as “Au-S-Ph” without discussing them in detail. Regarding the DFT, in the latter, only a single tetrahedral Au structure is attached to the single molecule and selection rules are not taken into account, which we find insufficient here (see Supp.Note 7) to capture accurately enough the low frequency molecular vibrations, that requires better representation of the Au facets.

While the reviewer also worries about subtracting the ERS background, no spurious bands can appear since only a best-fit model is subtracted. This is indeed a suitable way to treat such data since the molecular part should not be normalized by the electronic contribution (which comes from a completely separate process, and which in previous work was assumed (unsafely) to be 300K).

Finally, the comparison here of many different molecules, and using extremely challenging theoretical simulations, is the only way to truly test understanding. In particular, a key additional novelty here is the experimental data on single molecules (through picocavities) which shows that the influence of local environment. Indeed reviewer #3 emphasises how the large datasets that generate robust data here are crucial for exploiting the promise of low frequency Raman. This novelty is thus well suited for Nature Communications.

Reviewer #2:

1. In the introductory text, the authors stated *“...we push it to its low frequency limit, which is rarely utilized, ...”* Some of the references of this manuscript overlap with the current study in their scope, namely low frequency / THz SERS – for example, Inagaki et al, Curr Opin Electrochem, 2019 (ref 12), Inagaki et al JPCL 2017 (ref22), Sun et al JPCC 2022 (ref23). It would be better if the authors acknowledge these prior works and use them to better provide context to the current work.

> As suggested (and as in response to reviewer #1), we expand the comparison to previous work. A key difference is that we identify the contributions from intermolecular interaction to the THz modes, as well as using large datasets to separate vibrational and electronic contributions. This is now explicitly stated in paragraph 2 of the main text.

2. A strength of this study is the automated spectral acquisition from hundreds of particles. This allows more robust measurements and comprehensive sampling of molecular arrangements. In the manuscript and SI, mainly individual representative spectra were shown. One might view the selection of representative spectra as a form of ensemble averaging. Yet the strength of single particle measurements is to reveal individual variations, which are largely absent in the current presentation. This reviewer encourages the authors to also inform the reader about the variation and distribution of the spectra in the whole dataset.

> The selection of representative spectra is designed to differ from ensemble averaging in that it removes outliers and identifies commonality between spectra. As the reviewer notes, bulk measurements give ensemble averages whereas single-particle measurements allow the observation of individual variations. The selection process used here filters this variation in order to identify the spectra which are most common, and therefore representative of the sample. Ensemble averaging is unable to remove this variation. We now provide 30 individual spectra in each cluster in the SI Note 3 to give information about the variation, as requested, showing just how repeatable is our data.

3. In equation 1, the authors included both the Bose-Einstein (BE) and Fermi-Dirac (FD) distribution in modeling the spectral background. The BE distribution carries a T dependence which is used to infer the electronic temperature. The FD distribution, however, is represented using just a step function with no temperature dependence. Why?

> We apologize that this was not clear. The model in Eqn (1) is the Bose-Einstein distribution, which also has a step function between Stokes and antiStokes sides. We now clarify this as suggested, adding a suitable reference.

4. To analyze the low frequency SERS spectra, the authors adopted an unsupervised clustering algorithm implemented in the standard machine learning python library. Were full spectra used as input vectors to the clustering algorithm? It is clear the high frequency portions of the SERS spectra contain local vibrations from the thiol molecules. It would be interesting and informative to segment each SERS spectra into high frequency and low frequency portions and repeat the clustering analysis. Cross comparing the three cases (whole / high frequency / low frequency) would allow one to discuss the nature of spectral features more comprehensively. It would enable the authors to more clearly separate the spectral components in the low frequency range between molecular contributions and plasmonic metal contributions. Would one see both molecule-independent and molecule-dependent features?

> This a helpful clarification. The results in Fig.2 are already of segmented spectra - the input is the region below 200 cm^{-1} . Therefore, they do already enable the separation of molecular and plasmonic metal contributions to the spectra. The fact that they cluster well by molecule shows that differences in the THz part of the spectra are molecule dependent. We now clarify this in the text.

5. What is the meaning of n in the spring network model? It was not defined in the text.

> This was indeed omitted, and n is simply a positive integer $n=1,2,\dots$, which is now defined.

6. In the low frequency range, one could expect to observe two types of vibrations. One is the collective and global motion of individual molecules, the other is the external or lattice modes of molecules vibrating in an ordered lattice. The self-assembled monolayers (SAMs) of thiol molecules are known to exhibit good packing with a periodic lattice. One might argue that the DFT calculation should be done using a relaxed crystalline lattice. In this work VASP is used only to obtain the relaxed

structure, the spectral calculation is done using Gaussian, on isolated single molecular constructs. What is the justification for this choice? Wouldn't this choice of computation method entirely miss the lattice modes from the packed thiol molecules? There seems to be low frequency peaks, unaccounted for in the molecular calculation.

> Indeed the coupling of the molecules is an important point, and also raised by reviewer #3 below. We thus now significantly extended the DFT calculations to include multiple molecules, and show that the couplings are small enough that the mode behaviours are preserved (with neighbouring molecules either moving in-phase, or in antiphase), but is large enough that the modes are broadened. We added a new figure and discussion in the main text (see below).

7. Figure 3 a and b consistency. Figure 3a shows the measured spectra overlaid with Gaussian broadened DFT predictions. Figure 3b shows the measured spectra with a stick plot of simulated spectra. Please also show the stick plot predicted in the low frequency range.

> As suggested we now also add the sticks to the predicted spectra in the low frequency range.

8. Template stripping. The flat Au surface from template stripping is noted to be (111). Could the authors provide literature reference or data to support the identification of the Au film's orientation?

> This is a helpful question. Template-stripped Au is predominantly (111), which is shown by our electrochemical cyclic-voltammetry (Fig.R1 below). Distinct peaks are seen in Ag underpotential deposition (UPD) on Au (111) facets (see DOI: 10.1039/A906140A), that are not seen on evaporated Au. Indeed this data shows that much of but not all our surface is (111), which is why a large statistical evaluation is also important (as we show).

Fig. R1. Underpotential deposition of Ag on different Au electrodes. (a) Single crystal Au (111) electrode. (b) Template-stripped Au as used in the paper. (c) Evaporated Au. All data taken in 0.1M H₂SO₄ + 1mM Ag₂SO₄.

We also analyse room temperature STM measurements of BPT SAMs, which at high resolution show the expected hexagonal lattice that is known on (111) Au (Fig.R2 below). Average values from 10 repeat measurements match the expected SAM unit cell on the (111) Au surface.

Fig. R2. STM analysis of a BPT SAM on template stripped Au. (a) STM image, size 15nm, after background removal and Fourier filtering, $I_t = 0.3$ nA and $V_t = +0.3$ V. (b) 2D-FFT of STM image shown in (a). Red circles show regions of the 2D-FFT used to generate the Fourier filtered image shown in (c).

Reviewer #3:

1. The graphs in the article would benefit from further optimization. For instance, the experimental setup diagram does not provide a clear illustration on the role of some components in the developed THz SERS spectroscopy, such as the rationale behind using two volume holographic gratings (VHG) clean-up filters after the laser beam and three VHG notch filters in the collection path. Additionally, there is an inconsistency in the schematics above Fig. 1b and Fig. 1d, which needs further clarification. As shown in Fig. 1c, ν represents vibrational frequency, which is in general a positive number. Accordingly, Stokes and anti-Stokes Raman shifts should be expressed as $\omega_L - \nu$ and $\omega_L + \nu$, respectively. However, both the schematic above Fig. 1b and $\nu < 0$ assumed in Eq. 1 are different from such conventions. It is recommended that the author adjust the schematic diagram of Fig. 1b according to the generally accepted representation method to make it consistent with that of Fig. 1d.

> The reviewer helpful finds this inconsistency in Fig.1b which is now corrected. We also provide more explanation of the rationale for the setup in the Methods section, and took into account their suggestion for improving the clarity.

2. On page 3, the authors claim that THz vibrational modes strongly depend on molecular conformation - why is it the case particularly in THz region? The authors attribute the large FWHM of low-frequency Raman peaks to their local molecular conformation. Please provide further discussions to better illustrate this argument.

> The molecular conformation of molecules in self-assembled monolayers (SAM) is determined by the overall conformation of the molecule (not just short-range bonds) [see *Chem. Soc. Rev.* **41**, 2072 (2012)]. Contrary to larger-energy vibrations ($>500\text{cm}^{-1}$) which involve very localized motion of well-defined bonds, the vibrational modes in the THz region consist of delocalized vibrations over a large number of atoms in the molecule. As a result, the conformation of such molecules more strongly affects the THz spectrum. The large FWHM of low-frequency Raman peaks in the THz is the result of an increased density of vibrational modes in the THz region, when the molecule is attached to the metallic interface and coupled to neighbouring molecules, and the inhomogeneity due to different local environments at different positions in the gap (see point 10 below). We now explicitly discuss both these points in more detail in the main text.

3. In Fig. 2, the low-frequency SERS spectra of seven aromatic thiol molecules inside gold NPoMs were measured and compared. Different lengths and functional groups of these molecules result in varied effective compressibility. Please explain the physical meaning of "compressibility". In addition, the different junction conductance in these molecule-sandwiched gold NPoMs could also affect the measured SERS spectra, potentially due to electron transport through such ultrasmall plasmonic molecular nanogaps under light illumination. Please refer to *ACS nano*, 2018, 12(7): 6492-6503 for detailed information and make relevant discussions.

> This refers to the different stacking of molecules in the SAM (tilt angles and molecular spacing), that then influence how the very strongly attractive Van der Waals force compresses the SAM, and thus allows different THz vibrations. It is for this reason that simulations have to properly include the wider Au facets. Indeed we agree that molecular conductivity can affect the plasmons of the coupled system (as in the reference noted, and others), and it is possible it can also affect the vibrations. However our prior work [*Nature Comm* 8, 994 (2017)] showed that transport is so rapid for thiol-bound molecules that no vibrational effects are seen, though for higher barriers, stochastic switching is observed. No such effects are observed in the work here, in agreement with this prior work. We thus add a brief discussion as suggested.

4. Regarding Figs. 2a and 3a, it seems that the background-subtracted low-frequency modes in Fig. 3a are quite different from those original spectra in Fig. 2a. Please confirm whether the original data used for background subtraction in fig. 3a are the same as that in Fig. 2a; if not, please explain why different sets of data are used for comparison. Otherwise, the comparison with the simulated Gaussian-broadened Raman peaks would be meaningless. Please also add a paragraph in SI to illustrate the detailed procedures of background subtraction.

> We confirm that these are indeed the same spectra in Fig.2a and 3a, where the additional exponential background component has been removed. We describe this in the captions as suggested, and detail the procedures in SI note 2 and 4.

5. The authors mention that submerging the Au-coated glass in the molecular solution for 16-24h or 1h results in the adsorption of SAM on the surface of gold. In Table S3, the gap thicknesses of gold NPoMs set by each SAM is calculated based on the measured dark-field scattering spectra and an NPoM quasi-normal mode tool. It would be better to provide detailed explanations on the calculation process. Besides, in Fig. 3d, the blue shifts in the THz vibration modes as a function of the extracted gap thicknesses is explained by a spring network model. Please explain this model and its validity range and limitation.

> While the dependence of the dark-field plasmon mode position follows a clear dependence on NP diameter, gap size, gap contents and facet, the precise dependence does not follow a simple equation. We use full EM calculations which have been parametrised to allow the entire parameter space to be modelled. This is described in detail in DOI: 10.1021/acsphotonics.2c00116 (now added). The spring network model is simply a single effective spring which can exhibit multiple order modes of vibration, as described by Eqn(2), in order to give some intuition about the THz vibrations and compare to full DFT calculations. We thus clarify this, and the assumptions in the main text.

6. In Fig. 3a and Fig. 4, the authors fail to provide a clear explanation on the assignments of measured SERS peaks for the seven molecules in the THz region. Please add the assignments of molecular conformation or vibrational modes of the observed THz SERS peaks to better explore the origin of pico-cavity-induced low-frequency vibrations in the molecule-sandwiched NPoMs.

> We agree with the reviewer that this would be useful to add, showing the calculated vibrational modes to help characterize the nature of each one. We show now in the SI, the vibrational displacements of the two most intense vibrational modes for wavenumbers $< 200 \text{ cm}^{-1}$ ($< 6 \text{ THz}$) for each of the seven selected molecules anchored to two flat gold layers. We observe that these vibrational modes can be characterized by displacements of the molecular tail group along directions roughly parallel or perpendicular to the gold planar layers. Low-frequency modes in layered materials show similar vibrational patterns and they are classified as shear modes and layer-breathing modes, depending on whether the displacements of the atoms are mainly parallel or perpendicular to the layer plane [see *ACS Nano* **15**, 12509 (2021)]. Here, we use a similar nomenclature to classify the vibrational modes of molecules bound to the gold layers: shear modes when the atoms of the molecule move roughly parallel to the layers, and tapping modes (rather than layer breathing) when the atomic displacements are nearly perpendicular to the layers. Due to the differences in the way in which molecules assemble in SAMs compared to in layered crystals, particularly the Au-S-Ph angle in organic thiols on Au, the vibrational modes involve combinations of shear and tapping motions.

7. Figure 4 demonstrates that the molecular configuration can be effectively characterized by the low-frequency Raman modes. Interestingly, not only do the peaks below 200 cm^{-1} change with molecular vibrations, the peaks above 200 cm^{-1} also exhibit changes that can be utilized for configuration characterization. So what are the distinct advantages of utilizing the low-frequency vibrational signals for such characterization purposes? In

addition, it is unclear from Figs. 4d-4f which molecule is used to form the molecular junctions. Finally, it is better to demonstrate how the nanocavity and pico-cavity SERS spectra were extracted from the time series of SERS spectra.

> The particular interest of Fig.4 (now Fig.5), is that the linewidth of the picocavity-induced single-molecule spectra in the THz regime is much narrower, clearly showing the influence of different environments for each molecule. Such effects are not observed at higher frequencies (SI Fig.S9). In addition, since we have shown that the THz spectra give information about vibrations spanning the entire molecule, we gain key data to improve modelling of single molecule dynamics [shown to be challenging, eg see DOI: 10.1038/s41467-021-26898-1]. The nanocavity spectrum is a fit to the matrix of time series of SERS data, which ignores outlier points to identify the spectrum that is stable in time. The picocavity spectra are simply then produced by subtracting this nanocavity spectrum from the measured time series spectra, to visualize the time-varying SERS signals. We now clarify this in the caption, provide a specific description in the text, and clearly label the molecules used in Fig.5d-f.

8. The THz SERS spectra shown in Fig. 4 change significantly after the laser irradiation for 10 s. Is it solely due to the transformation from nanocavity to pico-cavity (enabling single-molecule SERS) or also caused by potential damage of molecular junctions? Additionally, the linewidths of low-frequency SERS peaks in the pico-cavity are much narrower than that in the nanocavity - the authors should make further discussions on this phenomenon in the context of (single-molecule) SERS.

> Indeed the dark-field spectra do not change over this time, which is good evidence that the changes only come from picocavities. We discuss this effect of narrower linewidth in point 10 below, along with additional discussions to better set the context in the main text.

9. The authors present a robust experimental approach and data processing method for obtaining clear Raman signals from molecule-sandwiched NPoM structures at low frequencies. While low-frequency Raman spectroscopy has been widely employed in the field of two-dimensional materials and molecular spectroscopy studies, it is important to highlight the unique contribution of this method and its significance when combined with plasmonic near-field enhancement and localization.

> Indeed we fully agree with the reviewer that combining robust data on many constructs along with low frequency Raman has great potential. We stress this now further in the conclusions.

10. The authors discuss in the main text that the low-frequency vibrational modes typically correspond to the complicated delocalized modes of molecular monolayers. However, the DFT simulation results shown in Fig. 3 are for single molecules, which may not accurately reflect the experimental results associated with molecular monolayers where multiple molecules are present in the cavities. Additionally, the spring network model seems to struggle to explain the findings of the pico-cavity with a molecular monolayer shown in Fig. 4c.

> We agree that this is a crucial point, since interactions between molecules can be important. Although we already discussed in the main text just how challenging extending these simulations are, we have now endeavoured to develop them enough to show their effects. In particular we find that molecular interactions do couple the molecules, but the character of the modes is retained although they split into mode sets (see Fig.R4 below). We feel this is extremely important, and thus add a new figure (Fig.4) to the main text along with discussions, and show the vibrational motions in each case.

Fig. R3: Collective effects in Raman spectra. Simulated Raman spectra for model SAMs based on (a) three BPT molecules (4) and (b) four MBN molecules (5), compared to the single molecules (b,d), when located between two flat gold layers. Simulated Raman peaks (solid lines) are Lorentzian broadened by 32 cm^{-1} for shaded curve, dashed lines connect related vibrational modes of single molecule and model SAMs.

Regarding application of the simple spring model to picocavities, the main challenge is understanding the perturbation introduced by the Au adatom in this constrained space on such low frequency modes. It is clear in Fig.5c that the modes remain in the same THz locations but likely there are 2 picocavities (compare to Fig.5d). We thus do not speculate further at this stage.

11. The present study utilizes plasmonic pico-cavities of ultrasmall mode volumes to probe molecule conformation through measuring low-frequency vibration modes below 200cm^{-1} . In a related study published in *J. Phys. Chem. Lett.* 2019, 10(16), 4692–4698, it was demonstrated that single plasmonic nanoparticles, such as gold nanorods with a much larger mode volume, could also serve as a SERS probe of molecular conformation and molecule-metal bonding configurations. Therefore, it would be interesting to compare this earlier study with the current one and elucidate specific advantages and necessity of employing plasmonic nanocavities and pico-cavities for similar purpose.

> This is an intriguing comparison for future work, since the JPCL paper noted discusses how the interaction with water can change the modes observed. In comparison to the JPCL work which averages over many thousands of molecules behaving more or less identically, the binding of molecules to metals evidently can be very heterogeneous (as noted in many papers over recent decades). The work here on nanocavities and picocavities is perhaps better considered as examining molecules in confined dimensions, and this work needs to be explored in subsequent investigations.

12. It has recently been demonstrated that the light-triggered conformation change of photoactive molecules confined inside a plasmonic nanocavity lead to reversible tuning of both SERS and SHG signals, <https://doi.org/10.1021/acs.nanolett.2c04988>. The author may also refer to this work and discuss how these low-frequency vibrational modes observed in the present nanocavities and pico-cavities affect their nonlinear emissions.

13. In the first paragraph of p6, some references should be added to support the assignments of Raman peaks.

> Indeed there are many interesting explorations of molecules in nanocavity environments, and we refer to the ones most pertinent to the study here on THz modes. It would indeed be intriguing if this research team can explore the effect of low frequency vibrations, however perhaps more important

is the irreversible changes they observe (which also we have reported in other papers) that involve the migration of Au adatoms that increase the facet size and retune the plasmon modes thus changing in/out-coupling. Here we explicitly avoid such effects by using CW average laser powers $<50\mu\text{W}$. We note that a key problem for nonlinear experiments is that optical forces scale with field strength, so to avoid Au facet changes for 100fs pulse excitation, $<1\text{nW}$ average powers are demanded. As suggested, the peak assignments made here from DFT are also referenced.

14. Please review and carefully check the citations in both manuscript and supporting information. For example, references 4 and 5 are identical in supporting information, and "Table S2" on page 3 of the text needs to be corrected to "Table S3".

> We thank the reviewer for spotting these, and amend as suggested.

REVIEWER COMMENTS

Reviewer #2 (Remarks to the Author):

The revised manuscript has address previous comments and shows improved scientific discussions. A few important issues persist that preclude a more favorable recommendation.

1.

As previously noted, the quality of this work is high. The central concern about this manuscript is its novelty. The authors tried to establish the distinction of this work from the previous works in a variety of aspects, including detailed discussions of molecular modes and DFT calculations, identifying intermolecular contributions, as well as machine learning approach to delineate vibrational and electronic contributions, etc. While all positive, in this reviewer's opinion, these aspects appear incremental and technical in their nature compared with existing reports and appear insufficient in warranting publication in Nature Communications.

Among the presented novelty claims, the claim of single molecule THz SERS from picocavities would be particularly interesting, if their single molecule nature can be adequately substantiated experimentally. Typically, THz range structural dynamics involve collective and global nuclear motions; showing such structural dynamics at the single molecule level would constitute a conceptual advancement. The authors claim single molecule THz vibrations in the manuscript, based on the laser induced, intensified fluctuating spectral response. The picocavity interpretation is reasonable, yet the single molecule nature of these spectral responses is not experimentally established. If one repeats the picocavity measurements with isotopically labelled molecules, then statistical correlations of observed isotope-specific vibrational frequencies should readily reveal whether the spectra indeed originate from single molecules. Can the authors provide such experimental evidence to establish the single molecule nature of their THz SERS from picocavities?

2.

The Fermi-Dirac background shown in Figure 1d seems to be problematic. The Bose-Einstein population factor is indeed incorporated into the standard expression of Raman scattering cross section. As a comparison, the authors stated that the intraband PL (background/lineshape) should follow a Fermi-Dirac distribution. What is the justification for this statement? There are certainly more factors, such as band structure and joint density of states, involved in the PL process and they could each have some energy dependence and spectral shape. It is not clear why it is reasonable to reduce all such factors to a spectrally flat constant and simple plot a Fermi-Dirac distribution as a spectral continuum background.

3.

The experimental spectral cut-off is at 5cm^{-1} , as shown in Figure 1(d) and experimental methods. Yet in all subsequent data analysis and discussions, the spectral cutoff is raised to 25cm^{-1} , losing much of the spectral access below 1THz. Why? Can the authors show the THz SERS spectra in their full spectral range?

4.

Related to point 3, the low frequency portion of the spectrum in Figure 1d resembles the spectral response of molecular liquids scaled by a Bose-Einstein factor of $[n_{\text{BE}}(\nu) + \theta(\nu)]$. Molecular liquids have dielectric relaxational dynamics reaching into the THz which show up in Raman scattering as a continuum spectral response. Such dielectric relaxation spectral response will converge to a finite value as frequency decreases and approaches zero, unlike the exponential function used in this work. One should be able to clearly distinguish the presence/absence of such low frequency spectral behavior in the accessible spectral region down to 5cm^{-1} . However, the 25cm^{-1} spectral cutoff used in Figure 2, 3 and throughout the rest of the analysis preclude such a discussion.

In summary, although this work is of high quality, its core novelty claim is yet to be convincingly established. Questions still remain with respect to several aspects of data analysis and scientific argumentation. In view of these issues, this reviewer recommends reconsideration after further major revision.

Reviewer #3 (Remarks to the Author):

Although the authors have made substantial improvements to the manuscript in response to the reviewers' comments, there still remain several issues to be addressed.

(1) Eq. 1 has intensity on the left-hand side, yet the nonlinear-optical susceptibility χ enters the equation as if the quantity on the left is the electric field. EF (enhancement factor) is usually written in terms of electric field, and thus the Raman intensity is proportional to its fourth power (which is correctly stated in page 2 though). Moreover, the authors claim that EF they use only accounts for the out-coupling efficiency at the Stokes frequency, ignoring the in-coupling efficiency at the fundamental frequency. Then, PC1 and PC2 in Fig. 2b are not defined properly, rendering the graph hardly useful.

(2) As the authors responded, the peak assignment results of the low-frequency Raman vibration modes in the terahertz region for those 7 molecules (added in Table S5) appear to be some inconsistent with the calculation results (as shown in Figure 3a). Based on the calculated Raman spectrum of molecule 5, it exhibits only one characteristic Raman peak in Figure 3a. However, there seems to be a discrepancy regarding two Raman peaks of 76 and 177 cm^{-1} as listed in Table S5. Besides, it is observed that the x-axis scale of the Raman peak in Figure 3a is not sufficiently clear to assign the Raman peaks. Therefore, it is recommended to optimize and adjust the x-axis scale to facilitate a more accurate interpretation.

(3) Additionally, I would suggest improving the writing of the manuscript and, in particular, to correct a few inaccurate and/or unclear sentences, such as "intra-band PL intensities should follow the excited electron population which is a Fermi-Dirac distribution", "light inelastically scatters from (bosonic) waves of the electron gas.", "non-zero skin depth of the NPoM plasmon that gives tight optical confinement".

We are delighted both reviewers emphasise our “*revised manuscript has address[ed] previous comments and shows improved scientific discussions*” and “*the quality of this work is high*” (reviewer 2) and that we “*have made substantial improvements to the manuscript in response to the reviewers’ comments*” (reviewer 3). We suitably answer all their questions below, in particular detailing the clear evidence for single molecule THz Raman.

Reviewer #2:

1. As previously noted, the quality of this work is high. The central concern about this manuscript is its novelty. The authors tried to establish the distinction of this work from the previous works in a variety of aspects, including detailed discussions of molecular modes and DFT calculations, identifying intermolecular contributions, as well as machine learning approach to delineate vibrational and electronic contributions, etc. While all positive, in this reviewer’s opinion, these aspects appear incremental and technical in their nature compared with existing reports and appear insufficient in warranting publication in Nature Communications.

Among the presented novelty claims, the claim of single molecule THz SERS from picocavities would be particularly interesting, if their single molecule nature can be adequately substantiated experimentally. Typically, THz range structural dynamics involve collective and global nuclear motions; showing such structural dynamics at the single molecule level would constitute a conceptual advancement. The authors claim single molecule THz vibrations in the manuscript, based on the laser induced, intensified fluctuating spectral response. The picocavity interpretation is reasonable, yet the single molecule nature of these spectral responses is not experimentally established. If one repeats the picocavity measurements with isotopically labelled molecules, then statistical correlations of observed isotope-specific vibrational frequencies should readily reveal whether the spectra indeed originate from single molecules. Can the authors provide such experimental evidence to establish the single molecule nature of their THz SERS from picocavities?

> We do not believe the advances here to be small. We show that even at THz frequencies, the molecular vibrations of the molecular layer can be understood from those of an individual molecule, and distinguished in a set of rather different molecules. We show that DFT calculations are difficult as they need to include the Au facets, but they can be developed to well account for the THz modes. As the reviewer agrees, single molecule THz modes are a key additional advance, enabled by picocavities.

Picocavities have now been extremely well established as tracking single molecules. Detailed evidence summarised in [42] includes multiple pieces of evidence originally presented nearly a decade ago [Science 2016, 354, 726]: (A) Measured picocavity formation energies match adatoms (~1 eV) for both Ag and Au (extracting more atoms requires more energy); (B) Adatom symmetry breaking alters the Raman selection rules as observed (since picocavity optical fields significantly vary along a single molecule); (C) Adatom-molecule coordination bonds seen in SERS are transient and fluctuate in time (see also detailed analysis in [Nat. Comm. 2021, 12, 6759]); (D) Adatoms only interact with a single neighbouring molecule, as observed from its vibrational wandering in time (though very occasionally two molecules with correlated wandering are seen); and (E) Simulations show only single atom features can create optical hot spots of volume below 1nm³ as experimentally supported by optomechanical thresholds and in STM.

While isotopic effects have been explored in such nanogaps with extreme optical confinement [eg see DOI 10.1073/pnas.1920091117], the picocavity vibrational peaks are shifted differently in each case (see papers above), as the particular configuration of Au adatom and molecule alters the coordination bond that changes the electron density across the molecule. This makes the conventional isotopic effect unavailable, but the observed new shifted lines (as in above Science paper) are equivalent. While further references could be added, these are all already clearly signposted in the brief review in [42], which is now briefly highlighted in the text.

2. The Fermi-Dirac background shown in Figure 1d seems to be problematic. The Bose-Einstein population factor is indeed incorporated into the standard expression of Raman scattering cross section. As a comparison, the authors stated that the intraband PL (background/lineshape) should follow a Fermi-Dirac distribution. What is the justification for this statement? There are certainly more factors, such as band structure and joint density of states, involved in the PL process and they could each have some energy dependence and spectral shape. It is not clear why it is reasonable to reduce all such factors to a spectrally flat constant and simple plot a Fermi-Dirac distribution as a spectral continuum background.

> Recent theory by Sivan et al [ACS Nano 15, 8724 ('21)] shows that the intra-band PL depends on electron occupation and indeed follows a Fermi-Dirac distribution (replotted in Fig.R1 below). Indeed other factors slightly change it, but because the bands here are mostly s-like, these are well known (and incorporated in the Sivan model). We thus add this key reference [33].

Fig.R1: Predicted intraband PL from [ACS Nano 15, 8724 ('21)], redrawn to match the data in Figure 1(d), and showing the Fermi-Dirac distribution.

3. The experimental spectral cut-off is at 5cm⁻¹, as shown in Figure 1(d) and experimental methods. Yet in all subsequent data analysis and discussions, the spectral cutoff is raised to 25cm⁻¹, losing much of the spectral access below 1THz. Why? Can the authors show the THz SERS spectra in their full spectral range?

> Additional features in this 3-25cm⁻¹ domain are not yet fully confirmed, and thus in the manuscript here we concentrate on higher wavenumbers. This is because the NPoM emits high angle light (>50° from sample normal) and ensuring this is suitably filtered using the VBGs is not trivial. As we note below, we are less interested in this exponential component in the present paper.

4. Related to point 3, the low frequency portion of the spectrum in Figure 1d resembles the spectral response of molecular liquids scaled by a Bose-Einstein factor of $[n_{BE}(\nu)+\theta(\nu)]$. Molecular liquids have dielectric relaxational dynamics reaching into the THz which show up in Raman scattering as a continuum spectral response. Such dielectric relaxation spectral response will converge to a finite value as frequency decreases and approaches zero, unlike the exponential function used in this work. One should be able to clearly distinguish the presence/absence of such low frequency spectral behavior in the accessible spectral region down to 5cm^{-1} . However, the 25cm^{-1} spectral cutoff used in Figure 2, 3 and throughout the rest of the analysis preclude such a discussion.

> The reviewer concentrates on the additional exponential background that we identify that is however not the main focus of this paper. To address their comment however, it is interesting that we do not see evidence of this component decreasing closer to the laser. It is unclear why dielectric relaxation in a liquid would be relevant in the samples examined here, however as suggested we now add a sentence mentioning this possibility.

Reviewer #3:

1. Eq. 1 has intensity on the left-hand side, yet the nonlinear-optical susceptibility χ enters the equation as if the quantity on the left is the electric field. EF (enhancement factor) is usually written in terms of electric field, and thus the Raman intensity is proportional to its fourth power (which is correctly stated in page 2 though). Moreover, the authors claim that EF they use only accounts for the out-coupling efficiency at the Stokes frequency, ignoring the in-coupling efficiency at the fundamental frequency. Then, PC1 and PC2 in Fig. 2b are not defined properly, rendering the graph hardly useful.

> We possibly misled the reviewer since a proportionality is used in equation (1) and we defined EF as the intensity enhancement, which corresponds to the square of the field enhancement. We thus now correct the notation as suggested to the conventional field enhancement. We do not include the in-coupling efficiency as it is constant, and explicitly note this now.

When performing principal components analysis, most variation in the data is by definition contained in the first two components known as PC1 and PC2. These are normalized and are therefore unitless. Additionally we cut the spectra to the region from $20\text{-}200\text{cm}^{-1}$ in order to compare the spectral shapes and not the absolute intensities (details are in SI note 3). Such PCA plots are typically used to show the effective separation of clustered data.

2. As the authors responded, the peak assignment results of the low-frequency Raman vibration modes in the terahertz region for those 7 molecules (added in Table S5) appear to be some inconsistent with the calculation results (as shown in Figure 3a). Based on the calculated Raman spectrum of molecule 5, it exhibits only one characteristic Raman peak in Figure 3a. However, there seems to be a discrepancy regarding two Raman peaks of 76 and 177cm^{-1} as listed in Table S5. Besides, it is observed that the x-axis scale of the Raman peak in Figure 3a is not sufficiently clear to assign the Raman peaks. Therefore, it is recommended to optimize and adjust the x-axis scale to facilitate a more accurate interpretation.

> We appreciate the reviewer's care in checking this data. The simulated Raman peak for molecule 5 indeed appears at 177cm^{-1} and is seen in Fig. 3b (red bar next to the left y-axis). This peak did not appear in Fig. 3a because the x-axis in this panel only extended to 175cm^{-1} , to help

concentrate on the low wavenumbers of interest. As suggested we now added extra labels to the x -scale in Fig.3a, and extend the x -axis. Note we also updated the theory curve for molecule #2 in this plot, to ensure that it is consistent with the full table in the SI.

3. Additionally, I would suggest improving the writing of the manuscript and, in particular, to correct a few inaccurate and/or unclear sentences, such as “intra-band PL intensities should follow the excited electron population which is a Fermi-Dirac distribution”, “light inelastically scatters from (bosonic) waves of the electron gas.”, “non-zero skin depth of the NPoM plasmon that gives tight optical confinement”.

> Recent theory shows that intra-band PL intensity should indeed follow the excited electron population. We cite this reference [33], and note it provides a key way to distinguish with ERS (however we stress this is not a key point of the current paper). We reference a review of extensive work in the 1980s which confirms that inelastic scattering off bosonic fluctuations of the electron gas in metals dominates [34], and now briefly mention this in the main text as suggested. We agree discussing the skin depth of NPoM plasmons is not such a key point here, and we thus now discuss in terms of optical field penetration, suitably citing [22]. These are helpful clarifications.

REVIEWER COMMENTS

Reviewer #2 (Remarks to the Author):

1.

In their rebuttal, the authors placed emphasis on aspects of (1) measuring molecular vibrations at low frequencies, (2) understanding the spectral features from individual, different molecules, and (3) including Au facets in the DFT calculations to account for THz modes.

In particular, Ikeda and coworkers have published a body of work on THz-SERS, for example, refs 19, 12, 20, 13 (chronologically ordered from 2017 to 2022) in this manuscript. Ref 20 serves as a useful reference for comparison with this manuscript. Ref 20 reported (1) observation of THz vibrational modes, (2) analysis on the electronic and vibrational origins of spectral features, and (3) facet dependent spectral responses in experimental observations.

In the aspects that the authors emphasized, key differences between ref 20 and this work appears to be: (1) this work reported measurement on more molecules; (2) associated with the more extensive spectral data, this work adopted machine learning methods in its analysis; (3) this work reported a more comprehensive computational effort in assigning low frequency vibrational modes.

This reviewer maintains the previous view that while these differences are certainly all positive, they have not brought forth substantially new insight or concept that changes our fundamental understanding for THz spectral responses in these plasmonic systems beyond recent literature.

2.

The claim of single molecule THz SERS is a potential point of new insight to be established, as noted in previous comments. The authors' notes on the superiority of the picocavities are appreciated.

The authors referred to their previous work on picocavities by Kamp et al 2020 for investigation of isotopic effects in similar experimental platform. In this paper, it was stated that "... suggesting fewer than ~100 molecules probed" (main text), and "We estimated 22 molecules in the nanogap from geometrical considerations, and later determined experimentally 20 molecules to be in the hotspot from measurements on isotopologues" (supplementary information).

This information extracted from Kamp et al 2020 addresses the previous question of this reviewer regarding if measured spectral response is of single molecule nature: the reported spectra are perhaps measured at the level of single molecule*s*; however, the readers would be misled if they interpret these results, following the text literally, as responses of *a single molecule*.

In view of the above, this reviewer is unable to support the publication of this manuscript in Nature Communications.

Reviewer #3 (Remarks to the Author):

The authors responded to the reviewer's comments by incorporating additional references and addressing specific issues raised in previous rounds of revisions. However, upon further reviewing the recent revisions, it has become evident that while there is increased discussion on innovativeness as proposed before, it does not fully explore its connection with the newly expanded claim regarding collective effects on THz SERS from picocavities. This aspect is indeed intriguing - therefore we recommend adjusting the article structure to prioritize a thorough discussion of a model for the THz

Raman background within your main text.

In addition, the discussion of the background is rather sloppy from the physical point of view. Reviewer's comments raising these issues were addressed rather cosmetically. To give a few examples, the authors insist on saying "intradband PL intensity follows the Fermi-Dirac distribution". Yet, intensity cannot follow a distribution; it can follow a dependence of a particular physical quantity on another, which is reflected in the said distribution. "Light scatters from bosonic waves" is another example, also mentioned in the previous round of revision. To the very least, a more accurate way of saying would be "bosonic quasiparticles", or "bosonic excitations". Further, "the background rises toward the laser" – toward the laser wavelength? "This background continuum arises from penetration of the NPoM plasmonic field into the metal due to its tight optical confinement" – The penetration alone cannot give rise to the background; instead, what happens is that owing to the light penetration, physical processes of the light-matter interaction in the metal are enabled (hot electron generation, electron-phonon scattering), and this in turn gives rise to the background. A few important words are skipped here and there, leaving the impression that this part of the manuscript did not get sufficient attention. Equation (1) is not "the Bose-Einstein expression" but merely includes it as one of the factors. "However, in the THz range it is necessary to select a specific fit function, appropriate to a physical model." – why is the THz range special in this context? A few more examples can be found throughout the manuscript, indicating the necessity of a proper proof-reading.

Overall, the content of the article is richer and the clarity of the manuscript has been improved to some extent. I would like to recommend a thoroughly revised version that has satisfactorily addresses both reviewers' comments for publication in Nature Communications.

Reviewer #2:

1. In their rebuttal, the authors placed emphasis on aspects of (1) measuring molecular vibrations at low frequencies, (2) understanding the spectral features from individual, different molecules, and (3) including Au facets in the DFT calculations to account for THz modes.

In particular, Ikeda and coworkers have published a body of work on THz-SERS, for example, refs 19, 12, 20, 13 (chronologically ordered from 2017 to 2022) in this manuscript. Ref 20 serves as a useful reference for comparison with this manuscript. Ref 20 reported (1) observation of THz vibrational modes, (2) analysis on the electronic and vibrational origins of spectral features, and (3) facet dependent spectral responses in experimental observations.

In the aspects that the authors emphasized, key differences between ref 20 and this work appears to be: (1) this work reported measurement on more molecules; (2) associated with the more extensive spectral data, this work adopted machine learning methods in its analysis; (3) this work reported a more comprehensive computational effort in assigning low frequency vibrational modes.

This reviewer maintains the previous view that while these differences are certainly all positive, they have not brought forth substantially new insight or concept that changes our fundamental understanding for THz spectral responses in these plasmonic systems beyond recent literature.

> The pioneering work cited by the reviewer is clearly very interesting and an important precedent to our work, and we further highlight these results in the new version of the manuscript. However, we would like to emphasize the points where we believe that our manuscript makes important new contributions:

- We are able to study experimentally the Raman signal emitted from molecules in picocavities. In our opinion, this is a particularly important result, both technically and fundamentally, as it opens a path to access the intrinsic lineshape of one (or a few) molecules and potentially to define a metric for molecular order. These results are discussed in more detail below.
- The extensive spectral features and machine learning analysis are not just useful as technical advances, but they also convey two important messages: (i) significant variations can be found between nominally identical NPoM structures also in the THz region, so that care is needed when analysing individual spectra (in contrast, Ref. 13 states *“This means that the vibrational*

characters in this low-frequency region are rather insensitive to the environmental conditions”), and (ii) despite these variations, it is still possible to classify the molecules.

- The more comprehensive computational effort in our manuscript leads in our opinion to at least two significant results: (i) a better understanding of the effect of the molecule-molecule interactions in THz SERS (in contrast with the general behaviour of high-energy vibrations that are much less sensitive to these interactions), and (ii) a clear demonstration of the strong influence of the local (atomistic) configuration of the gold surface on the chemical enhancement (clearer in the results in the Supporting information). Interestingly, this later conclusion is consistent with the experimental results in Ref. 19 which show clear changes in the Raman signal for (111) and (100) facets. Nonetheless, this similarity needs to be analysed with care because both works examine different phenomena: [19] focuses on the effect of the facet, while here we show: (i) the difference between flat gaps and tip-like configurations (the latter typical in SERS studies, such as Ref. [22]), and (ii) how small changes in the gap size can actually lead to different THz Raman spectra.
- We show that our systematic procedure removes the background and facilitates the analysis of the THz region of the Raman spectra. We emphasize that the background is removed here, which differs from e.g. Reference 20 where the background was used as a normalization factor to analyze the signal and to facilitate determining the temperature. Both procedures can be complementary to each other, but we highlight that the practicality of the method used in the current manuscript is illustrated in Figures 3,5, where we are able to identify the contribution from energetically similar vibrations in the THz spectral region.
- We do not only study additional molecules, but these molecules are more different than in the previous work, which allows us to capture additional physics. In reference 13, the same short molecule was used except for its termination, and the changes in the vibrational energies were attributed to the mass. In contrast, here the molecules are of very different length, and we emphasize the importance of the length of the molecule (size of the gap) in determining the vibrational energy. Thus, our paper introduces a new and complementary message to reference 13.

As a result, we strongly believe that our work includes important advances both technically and fundamentally. We believe that the importance of THz SERS will increase in the following years, and that the technical advances introduced here are an important contribution by themselves to facilitate progress in this area. Additionally, we also consider that the theoretical and experimental insights of this paper are significant advances towards elucidating, for example, the sensitivity of chemical effects on the environment and the origin of the line broadening and how to avoid such effects.

We aim now in our revised version to place the rather limited previous literature on THz SERS of this type of system in the right context and importance, and we better acknowledge its pioneering role in this context. We find our work novel and complementary in its understanding and characterization of THz vibrational modes in metallic nanostructures, and we are convinced that our encouraging results here will generate much further work on THz Raman in near future.

2. The claim of single molecule THz SERS is a potential point of new insight to be established, as noted in previous comments. The authors' notes on the superiority of the picocavities are appreciated. The authors referred to their previous work on picocavities by Kamp et al 2020 for investigation of isotopic effects in similar experimental platform. In this paper, it was stated that "... suggesting fewer

than ~100 molecules probed" (main text), and "We estimated 22 molecules in the nanogap from geometrical considerations, and later determined experimentally 20 molecules to be in the hotspot from measurements on isotopologues" (supplementary information).

This information extracted from Kamp et al 2020 addresses the previous question of this reviewer regarding if measured spectral response is of single molecule nature: the reported spectra are perhaps measured at the level of single molecule*s*; however, the readers would be misled if they interpret these results, following the text literally, as responses of *a single molecule*.

> We believe our previous response may have not clear enough regarding the use of isotopes. The previous work (Kamp et al 2020) considers both the signal from the whole nanocavity (where no picocavity is present) and from the picocavity. The estimation of the number of molecules mentioned by the reviewer, as well as the work with the isotopes, was performed for the nanocavities, not for the picocavities. In the case of the latter, the details of local environment are expected to affect the Raman signal more strongly than a change of isotope (explaining e.g. the changes with time in Figure 5c,f). Thus, we do not believe it would be fruitful to introduce different isotopes into picocavities, as it would not be possible to discriminate the changes due to the different isotopic mass from those due to the different environment.

We discuss next the number of molecules that the picocavity probes. We believe that the creation of sub-nanometric regions of strong fields ('hot spots') around atomic-sized protrusions has been firmly established in previous work by us and others. We already discussed the different lines of evidence in our previous reply, of which we can specially emphasize: (i) the changes in the relative weight of the peaks present in the Raman signal emitted from a picocavity (compared to a nanocavity), which can be explained by the extremely strong field gradients, and (ii) the achievement of optical mapping with submolecular resolution in STM configurations that exploit this extreme field confinement. Additionally, the results in the current manuscript further support the claim of probing a very small region: the recovery of narrow lines in the THz region, instead of the broad peaks in the nanocavity Raman spectrum, strongly suggests to us that the extreme field localization around picocavities allows for characterizing a few molecules (and thus avoids the broadening induced by averaging over many molecules). We do not see any other explanation for this behaviour.

At the same time, we accept that we cannot claim with full certainty that only one molecule is characterized. The hot spot may illuminate a very few molecules, instead of just one, and the local vibrational molecule-molecule interaction can introduce an effect on the measured signal from molecules that are not directly illuminated. Thus, we have weakened the claim of 'single molecule' in the revised text, and include "*or a few molecules*" everywhere, so that it is clear that the importance of the results resides in the extreme narrowing of the vibrations, which enables to identify and characterize the modes much better, rather than in the fact that a single may be probed by the picocavity. We emphasize however our belief that the measured picocavity spectra from one or a few molecules are an important result, as they indicate that the intrinsic lineshape is narrow and experimentally accessible, and that the broadening in the spectra measured from nanocavities is most likely due to an averaging effect.

Reviewer #3:

1. The authors responded to the reviewer's comments by incorporating additional references and addressing specific issues raised in previous rounds of revisions. However, upon further reviewing the recent revisions, it has become evident that while there is increased discussion on innovativeness as

proposed before, it does not fully explore its connection with the newly expanded claim regarding collective effects on THz SERS from picocavities. This aspect is indeed intriguing - therefore we recommend adjusting the article structure to prioritize a thorough discussion of a model for the THz Raman background within your main text.

> As discussed in the response to reviewer #2, the key aspect of our new modelling is to show that there is some influence of molecule-molecule interactions, but that experimentally picocavity THz spectra are much narrower suggesting they are not intrinsic. For the current paper, the THz signal is much more important than the Raman background (to avoid confusing readers), but we appreciate the reviewer suggestion and thus we better emphasize the THz Raman background in the new version of the manuscript.

2. In addition, the discussion of the background is rather sloppy from the physical point of view. Reviewer's comments raising these issues were addressed rather cosmetically. To give a few examples, the authors insist on saying "intradband PL intensity follows the Fermi-Dirac distribution". Yet, intensity cannot follow a distribution; it can follow a dependence of a particular physical quantity on another, which is reflected in the said distribution. "Light scatters from bosonic waves" is another example, also mentioned in the previous round of revision. To the very least, a more accurate way of saying would be "bosonic quasiparticles", or "bosonic excitations". Further, "the background rises toward the laser" – toward the laser wavelength? "This background continuum arises from penetration of the NPoM plasmonic field into the metal due to its tight optical confinement" – The penetration alone cannot give rise to the background; instead, what happens is that owing to the light penetration, physical processes of the light-matter interaction in the metal are enabled (hot electron generation, electron-phonon scattering), and this in turn gives rise to the background. A few important words are skipped here and there, leaving the impression that this part of the manuscript did not get sufficient attention. Equation (1) is not "the Bose-Einstein expression" but merely includes it as one of the factors. "However, in the THz range it is necessary to select a specific fit function, appropriate to a physical model." – why is the THz range special in this context? A few more examples can be found throughout the manuscript, indicating the necessity of a proper proof-reading.

> We thank the reviewer for their very careful reading of our manuscript. We have revised the full text carefully, and strongly tried to be very accurate in the description of the terminology used throughout the paper, including all the points above, and we trust that we have been able to solve all these issues.

Overall, the content of the article is richer and the clarity of the manuscript has been improved to some extent. I would like to recommend a thoroughly revised version that has satisfactorily addresses both reviewers' comments for publication in Nature Communications.

> We believe that the carefully revised version now fulfils the wish of the reviewer.